# Genome-wide CRISPR screening reveals genetic modifiers of mutant EGFR dependence in human NSCLC

Hao Zeng[1]*, Johnny Castillo-Cabrera[1], Mika Manser[2], Bo Lu[1], Zinger Yang[1], Vaik Strande[3], Damien Begue[3], Raffaella Zamponi[1], Shumei Qiu[2], Frederic Sigoillot[1], Qiong Wang[1], Alicia Lindeman[1], John S Reece-Hoyes[1], Carsten Russ[1], Debora Bonenfant[3], Xiaomo Jiang[1], Youzhen Wang[2], Feng Cong[1]*

[1]Chemical Biology and Therapeutics, Novartis Institutes for Biomedical Research, Cambridge, United States; [2]Oncology Disease Area, Novartis Institutes for Biomedical Research, Cambridge, United States; [3]Analytical Sciences and Imaging, Novartis Institutes for Biomedical Research, Basel, Switzerland

**Abstract** EGFR-mutant NSCLCs frequently respond to EGFR tyrosine kinase inhibitors (TKIs). However, the responses are not durable, and the magnitude of tumor regression is variable, suggesting the existence of genetic modifiers of EGFR dependency. Here, we applied a genome-wide CRISPR-Cas9 screening to identify genetic determinants of EGFR TKI sensitivity and uncovered putative candidates. We show that knockout of *RIC8A*, essential for G-alpha protein activation, enhanced EGFR TKI-induced cell death. Mechanistically, we demonstrate that RIC8A is a positive regulator of YAP signaling, activation of which rescued the EGFR TKI sensitizing phenotype resulting from *RIC8A* knockout. We also show that knockout of *ARIH2*, or other components in the Cullin-5 E3 complex, conferred resistance to EGFR inhibition, in part by promoting nascent protein synthesis through METAP2. Together, these data uncover a spectrum of previously unidentified regulators of EGFR TKI sensitivity in EGFR-mutant human NSCLC, providing insights into the heterogeneity of EGFR TKI treatment responses.

**\*For correspondence:**
life.zenghao@gmail.com (HZ);
feng.cong@novartis.com (FC)

## Introduction

Lung cancer is the leading cause of cancer-related mortality worldwide, with non-small-cell lung cancer (NSCLC) being the most common subtype (*Bray et al., 2018*; *Herbst et al., 2018*). Activating mutations in the kinase domain of epidermal growth factor receptor (EGFR) are present in about 10% to 40% of NSCLC patients, most frequently in-frame deletions in exon 19 (ex19 del) and a missense arginine-to-leucine mutation at codon 858 (L858R) (*Sharma et al., 2007*; *Pao and Chmielecki, 2010*). The approval and use of EGFR tyrosine kinase inhibitors (TKIs), such as erlotinib and gefitinib, have provided therapeutic breakthrough and achieved clinical success (*Kobayashi et al., 2005*; *Rosell et al., 2012*). However, the responses of EGFR-mutant NSCLC patients to EGFR TKIs are rarely complete with variable duration (*Kobayashi et al., 2005*; *Rosell et al., 2012*), suggesting that other factors could modulate the dependency of mutant EGFR and thus influencing EGFR TKI efficacy. Moreover, acquired resistance inevitably develops, leading to disease progression in almost all patients (*Pao and Chmielecki, 2010*).

A growing number of studies have been focusing on the understanding of mechanisms underlying EGFR TKI resistance, involving a variety of genetic and non-genetic alterations in signaling pathways and cell state (*Sequist et al., 2011*). Secondary EGFR on-target mutations, most frequently T790M mutation, account for about half of relapsed tumours with acquired resistance (*Kobayashi et al., 2005*; *Pao et al., 2005*; *Sequist et al., 2011*). Resistance can also result from reactivation of key

**eLife digest** Cancer is caused by cells growing and dividing uncontrollably as a result of mutations in certain genes. Many human lung cancers have a mutation in the gene that makes the protein EGFR. In healthy cells, EGFR allows a cell to respond to chemical signals that encourage healthy growth. In cancer, the altered EGFR is always on, which allows the cell to rapidly grow without any control, resulting in cancer.

One approach to treating these cancers is with drugs that block the activity of mutant EGFR. Although these drugs have been very successful, they do not always succeed in completely treating the cancer. This is because over time the cancer cells can become resistant to the drug and start forming new tumors. One way that this can happen is if random mutations lead to changes in other proteins that make the drug less effective or stop it from accessing the EGFR proteins. However, it is unclear how other proteins in cancer cells affect the response to these EGFR inhibiting drugs.

Now, Zeng et al. have used gene editing to systematically remove every protein from human lung cancer cells grown in the laboratory to see how this affects resistance to EGFR inhibitor treatment. This revealed that a number of different proteins could change how cancer cells responded to the drug. For instance, cells lacking the protein RIC8A were more sensitive to EGFR inhibitors and less likely to develop resistance. This is because loss of RIC8A turns down a key cell survival pathway in cancer cells. Whereas, cancer cells lacking the ARIH2 protein were able to produce more proteins that are needed for cancer cell growth, which resulted in them having increased resistance to EGFR inhibitors.

The proteins identified in this study could be used to develop new drugs that improve the effectiveness of EGFR inhibitors. Understanding how cancer cells respond to EGFR inhibitor treatment could help determine how likely a patient is to develop resistance to these drugs.

downstream signaling pathways originally inhibited by EGFR TKIs, such as PI3K-AKT and RAS-MEK-ERK pathways (*Chong and Jänne, 2013*; *Niederst and Engelman, 2013*; *Camidge et al., 2014*; *Rotow and Bivona, 2017*). For instance, resistance has been associated with amplification or activation of MET, HER2, FGFR and AXL, as well as PIK3CA mutation or loss of PTEN (*Engelman et al., 2006*; *Engelman et al., 2007*; *Sos et al., 2009*; *Takezawa et al., 2012*; *Zhang et al., 2012*; *Ware et al., 2013*). Moreover, activation of NF-kB and YAP signaling pathways also confers resistance to EGFR TKIs (*Bivona et al., 2011*; *Hsu et al., 2016*; *Chaib et al., 2017*). Non-genetic alterations, such as histological transformation from NSCLC to small cell lung cancer (SCLC) and epithelial-to-mesenchymal transition (EMT), have also been reported in relapsed tumors (*Sequist et al., 2011*). Furthermore, accumulating evidence has suggested that small subpopulations of cancer cells can evade lethal drug treatment by entering a drug tolerant 'persister' (DTP) state and serve as a founder for acquiring heterogeneous drug-resistance mechanisms upon long-term drug treatment (*Sharma et al., 2010*; *Ramirez et al., 2016*). Importantly, the mechanisms of EGFR TKI resistance elucidated to date can encompass multiple mechanisms simultaneously in one patient or even tumour, creating significant obstacles for designing better treatment strategies for patients (*Pao and Chmielecki, 2010*; *Chong and Jänne, 2013*). Moreover, a number of EGFR TKI-resistant tumours lack known resistance mechanisms (*Sequist et al., 2011*), suggesting the existence of previously unrecognized mediators of EGFR TKI efficacy.

In order to better understand the incomplete response of EGFR-mutant NSCLC to EGFR TKI as well as identify prospective mechanisms of resistance, we conducted a genome-scale CRISPR-Cas9 genetic screen in a human NSCLC cell line harbouring activating EGFR mutation. By generating EGFR TKI resistance, we identified a number of genes, when deleted, enhanced or reduced EGFR TKI sensitivity and consequently prevented or accelerated development of EGFR TKI resistance, respectively.

# Results

## Genome-wide CRISPR-Cas9 screening reveals genetic modifiers of EGFR TKI sensitivity in EGFR-mutant NSCLC

HCC827, a very commonly used EGFR-mutant NSCLC cell line, harbors ex19 del in EGFR and is highly sensitive to EGFR TKI treatment. Generation of a dose-response curve upon 3 days' erlotinib exposure yielded the IC$_{50}$ 25.6 ± 3.6 nM (*Figure 1A*). However, even with high dose of erlotinib (1 µM and above), about 30% of the cells could still survive this initial pulse (*Figure 1A*). Examination of the cell proliferation in the presence of DMSO control or 1 µM erlotinib over a 30 day period using IncuCyte demonstrated that HCC827 cells were still able to proliferate at a low rate when exposed to high dose of erlotinib (*Figure 1B*). Additionally, colony formation assay also confirmed the existence of a small fraction of viable cells after 9 days' erlotinib treatment, referred to as 'drug tolerant persister' (DTP) cells (*Figure 1C*). Thereafter, the DTP cells commenced cell proliferation in the presence of erlotinib, yielding colonies of cells referred to as 'drug tolerant expanded persister' (DTEP) cells or drug resistant cells (*Figure 1C*). These data suggested that EGFR inhibition in the cultured cells mimics clinical observations of the incomplete response and/or innate resistance to EGFR TKI treatment, allowing the assay window to screen for mediators of EGFR TKI sensitivity. We aimed to systematically identify genetic modifiers that regulate the response of EGFR-mutant NSCLC cells to EGFR TKI treatment by applying a genome-scale CRISPR-Cas9 loss-of-function screening approach (*Figure 1D*). We introduced a pooled lentiviral single guide RNA (sgRNA) library targeting 18,360 genes (five sgRNAs per gene) into HCC827 cells with constitutive Cas9 expression (HCC827-Cas9) and treated these cells with DMSO or erlotinib. We intentionally applied high dose of erlotinib (1 µM) to allow the survival of a small subpopulation of DTP cells and development of drug resistance in the long-term. This strategy ensured that genes whose deletion synergize with or confer resistance to erlotinib could be negatively or positively selected from the screen, respectively, following erlotinib treatment compared to DMSO treatment. Three weeks post-treatment, cells were harvested and subjected to next-generation sequencing (NGS) to identify differential sgRNA representation between DMSO and erlotinib treated populations (*Figure 1D* and *Figure 1—figure supplement 1A–C*). Differential sgRNA representation was evaluated in the form of log2 fold change between the erlotinib- and DMSO-treated samples. A robust z score was calculated using the median and mean-absolute deviation for the calculated fold changes across the entire sgRNA library. To provide a qualitative assessment of the screen performance, we plotted the *P* values calculated by the redundant small interfering RNA (siRNA) activity (RSA) test, representing the probability of a gene hit based on the collective activities of multiple sgRNAs per gene, against Q1- and Q3-based z scores (*Figure 1E–F*).

To identify high-confidence negatively selected hits, we used a stringent RSA $\leq -4$ and Q1 z-score $\leq -1$ threshold. This analysis identified 35 genes, the inactivation of which caused sensitization to erlotinib in HCC827 cell line (*Figure 1E*, *Figure 1—figure supplement 1D*, *Supplementary file 1* and *2*). FGFR1 and YAP1, two known mediators of EGFR TKI resistance in EGFR-mutant NSCLC (*Ware et al., 2013*; *Hsu et al., 2016*; *Chaib et al., 2017*; *Ghiso et al., 2017*), were among these hits. Mapping the 35-gene set on the STRING protein-protein interaction network and the Reactome database generated clusters consisting of heparan sulfate metabolism, GPCR/G-protein signaling, Hippo-YAP signaling pathway, as well as components of the SAGA/transcriptional complex (*Figure 1G* and *Figure 1—figure supplement 1F*), providing potential synthetic lethal partners with EGFR TKI in EGFR-mutant NSCLC. A less stringent threshold (RSA $\leq -3$ and Q1 z-score $\leq -1$) generated a larger list of 122 genes whose loss sensitized HCC827 cells to erlotinib treatment (*Supplementary file 1* and *2*).

Similarly, we applied a stringent RSA $\leq -4$ and Q3 z-score $\geq 1$ threshold to identify 47 genes whose inactivation conferred resistance to erlotinib in HCC827 cell line (*Figure 1F*, *Figure 1—figure supplement 1E*, *Supplementary file 1* and *2*). A less stringent threshold (RSA $\leq -3$ and Q3 z-score $\geq 1$) expanded the list to 171 genes (*Supplementary file 1* and *2*), including negative regulators of YAP signaling such as NF2/Merlin and SAV1, further warranting the importance of YAP signaling in mediating EGFR TKI resistance in lung cancer. Protein network and pathway analyses of the 47 positively selected hits enriched for PI3K-mTOR signaling (such as PTEN, TSC1 and TSC2) and RAS-MAPK signaling (such as NF1, SPRY2 and LZTR1) pathways (*Figure 1H* and *Figure 1—figure supplement 1G*), two well-established modes of resistance to EGFR TKI in EGFR-mutant NSCLC

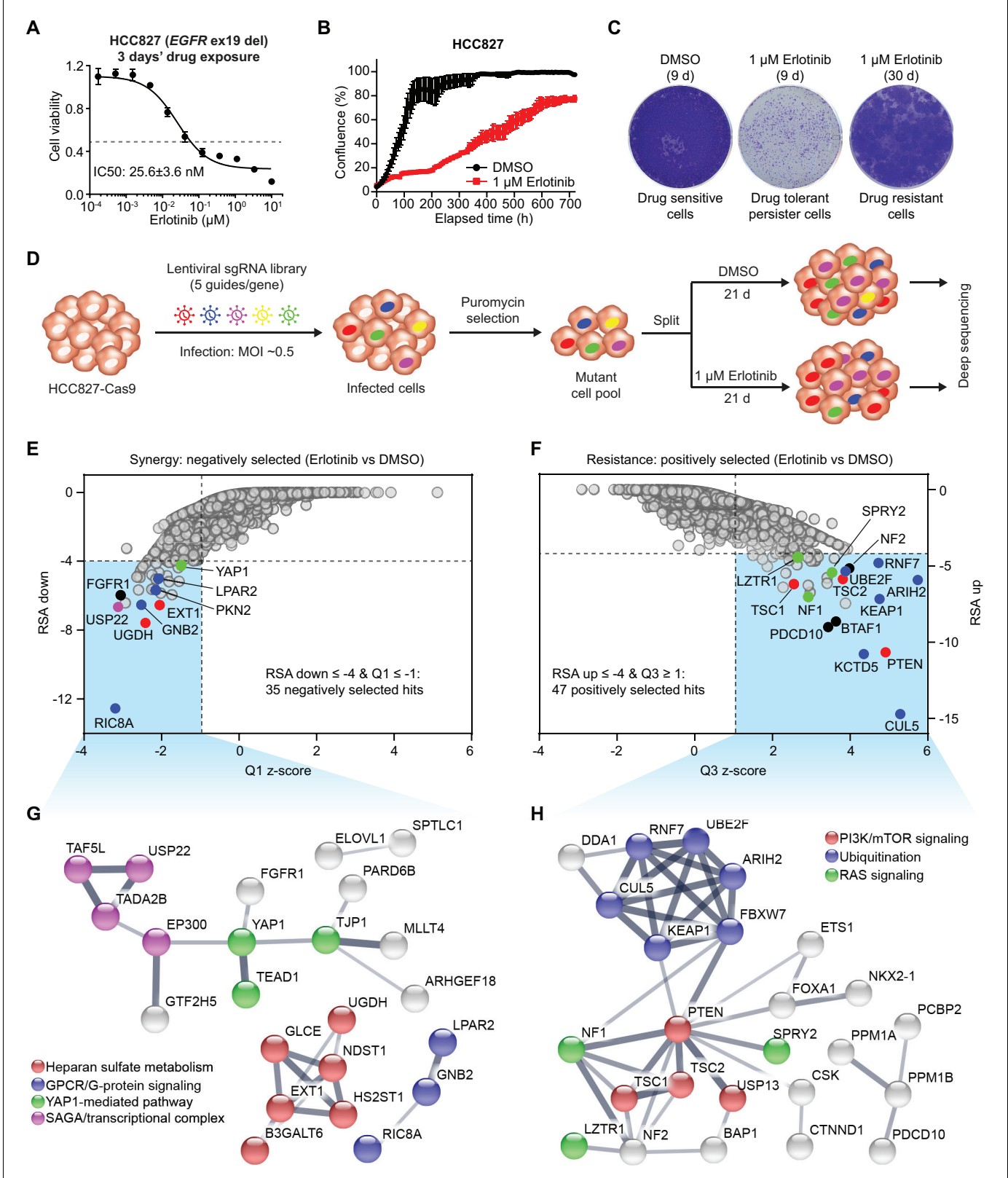

**Figure 1.** Genome-wide CRISPR-Cas9 screening identifies determinants of EGFR-TKI sensitivity in EGFR-mutant NSCLC. (**A**) Cell viability assessment by CellTiter-Glo assay of HCC827 cells treated with serial dilutions of erlotinib for 72 hr. Error bars represent mean ± standard deviation (SD); n = 4. (**B**) Kinetic cell proliferation assay monitored by IncuCyte for HCC827 cells cultured in the presence of DMSO control or 1 μM erlotinib over a 30 day

*Figure 1 continued on next page*

**Figure 1 continued**

period. (**C**) Crystal violet staining colony formation assay of HCC827 cells treated with DMSO or 1 µM erlotinib for the indicated days. (**D**) Schematic outline of the genome-wide CRISPR-Cas9 screening workflow in HCC827 cells. (**E**) Scatterplot depicting gene level results for erlotinib negatively selected hits in the CRISPR screen. A number of representative hits are shown in color. (**F**) Scatterplot depicting gene level results for erlotinib positively selected hits in the CRISPR screen. A number of representative hits are shown in color. (**G**) STRING protein network of the 35 negatively selected hits as defined in (**E**). The nodes represent indicated proteins, and colored nodes highlight proteins enriched in certain signaling pathways. The edges represent protein-protein associations, and the line thickness indicates the strength of data support. The minimum required interaction score was set to default medium confidence (0.4), and the disconnected nodes were removed from the network. (**H**) STRING protein network of the 47 positively selected hits as defined in (**F**).

The online version of this article includes the following figure supplement(s) for figure 1:

**Figure supplement 1.** CRISPR-Cas9 screening reveals genetic determinants of EGFR-TKI sensitivity.

(*Pao and Chmielecki, 2010*; *Chong and Jänne, 2013*; *Niederst and Engelman, 2013*; *Rotow and Bivona, 2017*), validating the good performance of our CRISPR screen. Moreover, several genes encoding proteins involved in protein ubiquitination and degradation pathway were positively selected from the screen (*Figure 1F and H*). Among them, we identified *KEAP1* and *FBXW7*, the loss of which have previously been shown to confer resistance to EGFR TKI treatment in EGFR-mutant NSCLC cells (*Krall et al., 2017*; *Ye et al., 2017*). Interestingly, components of the Cullin 5 (CUL5)-RING E3 ligase (CRL5) complex (such as *CUL5*, *RNF7* and *UBE2F*) as well as *ARIH2*, an Ariadne family RING-in-Between-RING (RBR) E3 ligase working together with CRL5, were amongst the positively selected hits, suggesting their previously unrecognized role in mediating EGFR TKI resistance in lung cancer. Together, our genome-wide CRISPR-Cas9 loss-of-function genetic screen successfully revealed both known and potential synthetic lethal vulnerabilities with EGFR TKI as well as modes of resistance in EGFR-mutant NSCLC.

## Validation of candidate mediators of EGFR TKI sensitivity

Next, we set out to validate a number of selected hits from the primary screen. We first focused on the synergy hits including *YAP1*, *USP22* and GPCR/G-protein signaling-related *LPAR2*, *GNB2*, *PKN2* and *RIC8A*, by selecting one sgRNA for each gene and monitoring the gene deficiency and erlotinib efficacy by western blots (*Figure 2A*). Long-term colony formation assays confirmed that all sgRNAs targeting these genes strongly sensitized HCC827 cells to both erlotinib and gefitinib treatment (*Figure 2B–C*). The synthetic lethal phenotype was similarly observed in another EGFR-mutant NSCLC cell line NCI-H3255 that expresses EGFR-L858R (*Figure 2—figure supplement 1A–C*). For validation of resistance hits, we intentionally selected individual sgRNAs targeting novel mediators of erlotinib resistance (*ARIH2*, *CUL5*, *RNF7*, *KCTD5*, *PDCD10* and *BTAF1*) (*Figure 2D*) and monitored their ability to confer resistance to EGFR TKIs by performing long-term colony formation assays. Our results confirmed that all sgRNAs demonstrated significant resistance to both erlotinib and gefitinib treatment in HCC827 cells (*Figure 2E–F*) as well as in NCI-H3255 cells (*Figure 2—figure supplement 1D–F*). Together, the validation study corroborated our CRISPR screen findings, providing confidence to investigate additional hits generated from our CRISPR screen (*Supplementary file 1* and *2*).

*LPAR2*, one of our strongest synergy hits, is a member of the lysophosphatidic acid receptor (LPAR) family. LPARs, consisting of LPAR1-6, are membrane-bound G-protein-coupled receptors (GPCRs) (*Yung et al., 2014*). Inspection of sgRNAs targeting all six LPAR members demonstrated that deletion of *LPAR1/2/3* tends to have synergistic effect with erlotinib treatment in HCC827 cells, although only *LPAR2* passed our stringent hit selection criteria (*Figure 2—figure supplement 2A*). We then asked whether pharmacologically targeting LPARs with LPAR antagonists could synergize with EGFR TKI to prevent drug resistance. Indeed, colony formation assays in HCC827 and NCI-H3255 cells demonstrated that LPAR antagonists, such as Compound 2 (LPAR1 selective) (*Qian et al., 2012*), AM095 (LPAR1 selective) (*Swaney et al., 2011*) and Ki16425 (targeting LPAR1-3) (*Ohta et al., 2003*), synergized with erlotinib to inhibit clonogenic cell growth (*Figure 2—figure supplement 2B–C*). These results nominate LPAR antagonists might, in principle, be combined with EGFR inhibition to delay resistance occurrence.

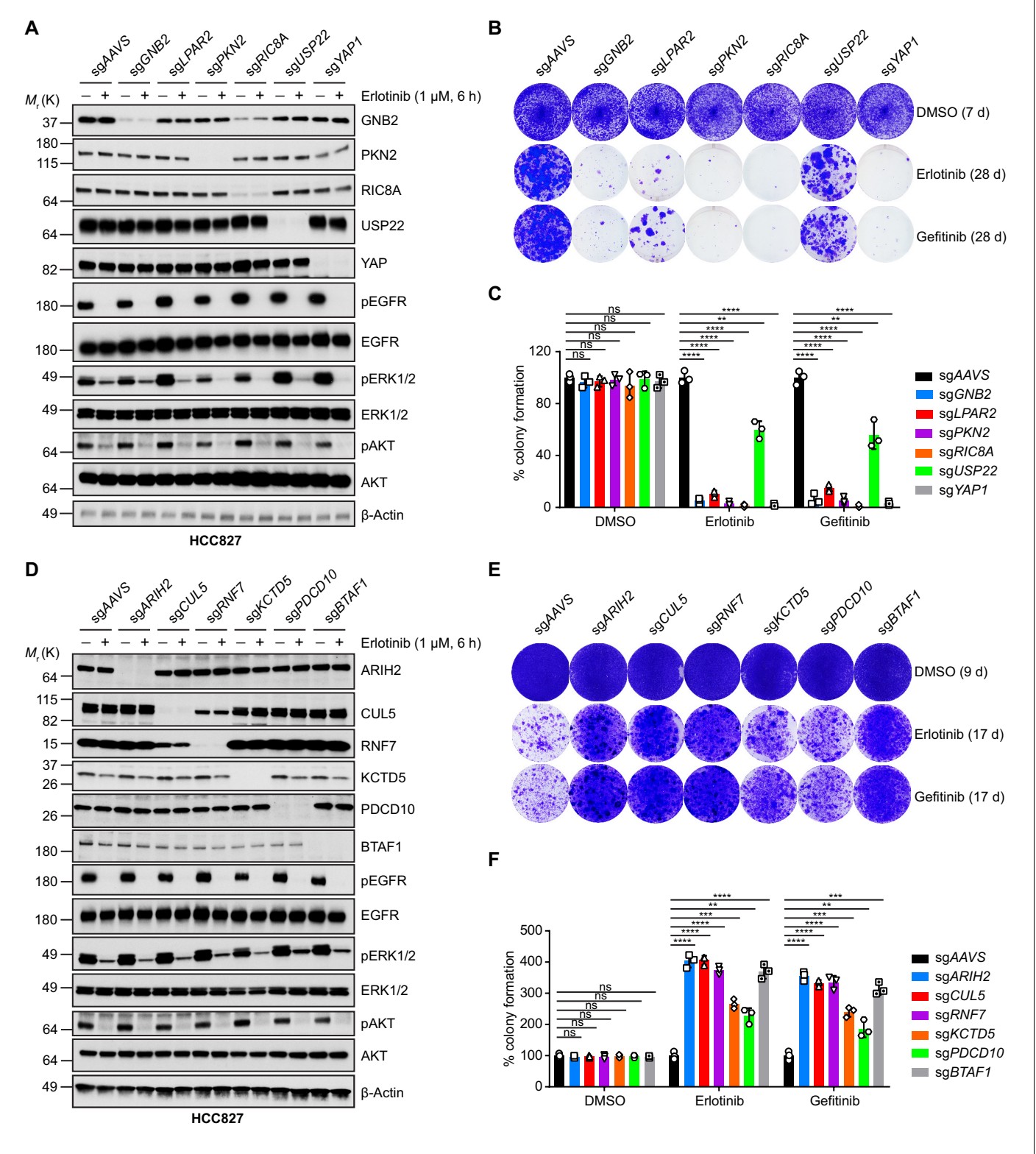

**Figure 2.** Validation of selected hits by individual knockout in HCC827 cells. (**A**) Immunoblots of indicated proteins in cells treated with DMSO or erlotinib (1 μM) for 6 hr to confirm specific knockout of negatively selected hits and on-target inhibition of EGFR pathway by erlotinib treatment. β-Actin was used as a loading control. Individual knockout cell lines were generated by lentivirus-mediated expression of sgRNA targeting indicated genes in HCC827 cells with constitutive Cas9 expression. (**B**) Crystal violet staining colony formation assay of indicated HCC827 cell lines treated with DMSO, erlotinib (1 μM), or gefitinib (1 μM). (**C**) Quantification of colony formation in (**B**), shown as percentage of the sg*AAVS* sample. Mean (three biological

*Figure 2 continued on next page*

*Figure 2 continued*

replicates) ± standard deviation (SD) is shown. (**D**) Immunoblots of indicated proteins in cells treated with DMSO or erlotinib (1 µM) for 6 hr to confirm specific knockout of positively selected hits and on-target inhibition of EGFR pathway by erlotinib treatment. β-Actin was used as a loading control. (**E**) Crystal violet staining colony formation assay of indicated HCC827 cell lines treated with DMSO, erlotinib (1 µM), or gefitinib (1 µM). (**F**) Quantification of colony formation in (**E**), shown as percentage of the sg*AAVS* sample. Mean (three biological replicates) ± SD is shown. Statistical significance was tested using unpaired two-tailed t test (**C and F**); *p<0.05, **p<0.01, ***p<0.001, ****p<0.0001; ns, not significant.

The online version of this article includes the following source data and figure supplement(s) for figure 2:

**Source data 1.** Raw data from *Figure 2*.
**Figure supplement 1.** Validation of selected hits by individual knockout in NCI-H3255 cells.
**Figure supplement 2.** Targeting LPARs sensitizes EGFR-mutant NSCLC cells to EGFR inhibition.

## *RIC8A* loss is synthetic lethal with EGFR inhibition in EGFR-mutant NSCLC cells

Among the novel synergy hits, we further focused on the most prominent hit, *RIC8A*, and decided to confirm its role in cell survival upon EGFR inhibition in more details. RIC8A functions as a biosynthetic chaperone and guanine nucleotide exchange factor (GEF) for a subset of G protein α subunits (*Nishimura et al., 2006*; *Gabay et al., 2011*; *Chan et al., 2013*; *Kant et al., 2016*). Previous study has shown that *RIC8A* deficiency leads to loss of Gα subunits, and consequently, Gβγ dimers dissociated from Gα subunits could be recognized by KCTD5 for degradation (*Chishiki et al., 2013*; *Boularan et al., 2015*; *Brockmann et al., 2017*). First, we deleted *RIC8A* in HCC827 cells using two independent sgRNAs. Western blot analysis confirmed *RIC8A* knockout efficiency and the consequent effect on the protein abundance of Gαq (GNAQ) and Gβ2 (GNB2) subunits as well as the on-target inhibition of EGFR and its downstream signaling upon erlotinib treatment (*Figure 3A*). We then examined whether loss of *RIC8A* affects the overall EGFR TKI response in HCC827 cells by generating a dose-response curve. Cells deficient in *RIC8A* exhibited enhanced sensitivity to both erlotinib and gefitinib (*Figure 3B* and *Figure 3—figure supplement 1A*). Consistently, *RIC8A* depletion dramatically accelerated apoptosis induction by erlotinib, represented by the increased caspase 3/7 activity (*Figure 3C*). To further explore the role of RIC8A in a long-term drug treatment associated with acquired drug resistance, we performed the colony formation assays in the absence or presence of erlotinib and gefitinib. Our results demonstrated that loss of *RIC8A* had no effect on the basal proliferation of HCC827 cells while dramatically suppressed erlotinib- and gefitinib-resistant colony formation (*Figure 3D–E*), which could be rescued by overexpression of CRISPR/Cas9-resistant RIC8A (*Figure 3—figure supplement 1B*). Similar effects were also observed in two additional NSCLC cell lines with the same EGFR ex19 del, HCC4006 and PC9 (*Figure 3—figure supplement 1E–L*). Moreover, we also showed that loss of *RIC8A* sensitized cells to EGFR TKI treatment and efficiently prevented erlotinib- and gefitinib-resistant colony formation in NCI-H3255 cell line harboring EGFR-L858R mutation (*Figure 3F–3J* and *Figure 3—figure supplement 1C*).

EGFR T790M 'gatekeeper' mutation is a major resistance mechanism in EGFR-mutant NSCLC patients in response to first-generation EGFR TKI treatment (*Pao et al., 2005*; *Sequist et al., 2011*). Third-generation EGFR TKIs, such as EGF816 (*Jia et al., 2016*; *Lelais et al., 2016*) and the FDA-approved agent osimertinib (*Jänne et al., 2015*), can bind to and inhibit mutant EGFR with and without the T790M mutation. However, the third-generation EGFR TKIs also produce a partial response followed by progression and acquired resistance (*Rotow and Bivona, 2017*). To investigate the potential role of RIC8A in mediating response to third-generation EGFR TKI, we deleted *RIC8A* in NCI-H1975 cells, which harbor EGFR-L858R and T790M mutations and thus insensitive to erlotinib (*Figure 3K* and *Figure 3—figure supplement 1D*). Strikingly, loss of *RIC8A* in NCI-H1975 cells dramatically enhanced the growth suppressive effect (*Figure 3L*), promoted apoptosis (*Figure 3M*), and attenuated development of resistance (*Figure 3N and O*) upon EGF816 treatment. Moreover, loss of *RIC8A* exhibited no effect on the EGFR TKI sensitivity in EGFR wild-type (EGFR-WT) NSCLC cell lines (A549, NCI-1299 and NCI-H460) or normal human bronchial epithelial cell line BEAS-2B (*Figure 3—figure supplement 2A–P*). Taken together, these data suggest that loss of *RIC8A* is synthetic lethal with EGFR inhibition across a panel of NSCLC cell lines with various EGFR mutations, representing a general mechanism.

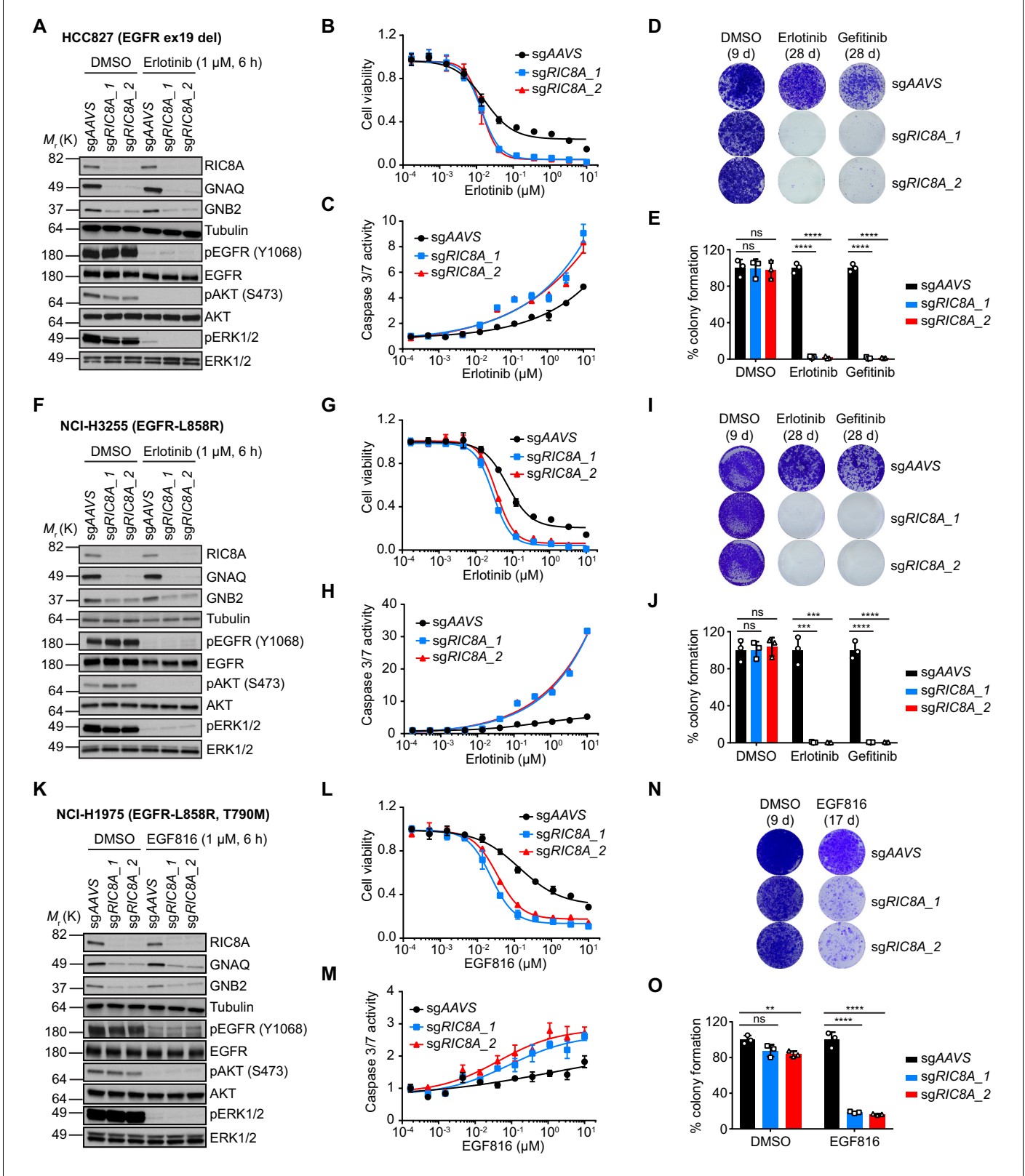

**Figure 3.** RIC8A depletion causes synthetic lethality with EGFR-TKI in EGFR-mutant NSCLC. (**A**) Immunoblots of indicated proteins in control (sg*AAVS*) or RIC8A knockout (sg*RIC8A*) HCC827 cells treated with DMSO or erlotinib (1 μM) for 6 hr to confirm specific knockout of RIC8A and on-target inhibition of EGFR pathway by erlotinib treatment. Tubulin was used as a loading control. (**B**) Cell viability assessment by CellTiter-Glo assay of HCC827

*Figure 3 continued on next page*

*Figure 3 continued*

cells treated with serial dilutions of erlotinib for 72 hr. Error bars represent mean ± standard deviation (SD); n = 4. (C) Activated caspase 3/7 measurement of HCC827 cells treated with serial dilutions of erlotinib for 24 hr. Error bars represent mean ± SD; n = 4. (D) Crystal violet staining colony formation assay of indicated HCC827 cell lines treated with DMSO, erlotinib (1 µM), or gefitinib (1 µM). (E) Quantification of colony formation in (D), shown as percentage of the sg*AAVS* sample. Mean (three biological replicates) ± SD is shown. (F) Immunoblots of indicated proteins in control or RIC8A knockout NCI-H3255 cells treated with DMSO or erlotinib (1 µM) for 6 hr to confirm specific knockout of RIC8A and on-target inhibition of EGFR pathway by erlotinib treatment. Tubulin was used as a loading control. (G) Cell viability assessment by CellTiter-Glo assay of NCI-H3255 cells treated with serial dilutions of erlotinib for 72 hr. Error bars represent mean ± SD; n = 4. (H) Activated caspase 3/7 measurement of NCI-H3255 cells treated with serial dilutions of erlotinib for 24 hr. Error bars represent mean ± SD; n = 4. (I) Crystal violet staining colony formation assay of indicated NCI-H3255 cell lines treated with DMSO, erlotinib (1 µM), or gefitinib (1 µM). (J) Quantification of colony formation in (I), shown as percentage of the sg*AAVS* sample. Mean (three biological replicates)± SD is shown. (K) Immunoblots of indicated proteins in control or RIC8A knockout NCI-H1975 cells treated with DMSO or EGF816 (1 µM) for 6 hr to confirm specific knockout of RIC8A and on-target inhibition of EGFR pathway by EGF816 treatment. Tubulin was used as a loading control. (L) Cell viability assessment by CellTiter-Glo assay of NCI-H1975 cells treated with serial dilutions of EGF816 for 72 hr. Error bars represent mean ± SD; n = 4. (M) Activated caspase 3/7 measurement of NCI-H1975 cells treated with serial dilutions of EGF816 for 24 hr. Error bars represent mean ± SD; n = 4. (N) Crystal violet staining colony formation assay of indicated NCI-H1975 cell lines treated with DMSO or EGF816 (1 µM). (O) Quantification of colony formation in (N), shown as percentage of the sg*AAVS* sample. Mean (three biological replicates) ± SD is shown. Statistical significance was tested using unpaired two-tailed t test (E, J and O); *p<0.05, **p<0.01, ***p<0.001, ****p<0.0001; ns, not significant. The online version of this article includes the following source data and figure supplement(s) for figure 3:

**Source data 1.** Raw data from *Figure 3*.
**Figure supplement 1.** RIC8A loss is synthetic lethal with EGFR-TKI in EGFR-mutant NSCLC.
**Figure supplement 2.** RIC8A loss exhibits no effect on EGFR TKI sensitivity in EGFR-WT NSCLC cells or normal human bronchial epithelial cells.

## Loss of *RIC8A* attenuates YAP signaling to synergize with EGFR inhibition in EGFR-mutant NSCLC cells

Next, we aimed to understand the molecular mechanism through which *RIC8A* loss synergizes with EGFR inhibition. Gα protein coupled GPCR signaling has been well characterized to modulate activities of Rho/Rac GTPase, which in turn lead to actin cytoskeleton remodeling, consequently regulating YAP signaling via both LATS1/2-dependent and LATS1/2-independent mechanisms (*Yu et al., 2012*; *Ma et al., 2019*). Given the important roles of RIC8A in mediating Gα activation, we hypothesized that RIC8A is a positive regulator of YAP signaling. To test this hypothesis, we first deleted *RIC8A* in HEK293A cells using two independent sgRNAs (*Figure 4A*) and verified that loss of *RIC8A* significantly decreased YAP reporter (GTIIC-GFP) activity (*Figure 4B* and *Figure 4—figure supplement 1A*), reduced expression of classical YAP target genes (*ANKRD1*, *CTGF*, and *CYR61*) (*Figure 4C*), and suppressed YAP-dependent growth of HEK293A cells (*Figure 4—figure supplement 1D*). Furthermore, these defects resulting from *RIC8A* loss can be overcome by overexpression of the constitutively active YAP (YAP-5SA) (*Figure 4—figure supplement 1B–D*). Together, these data strongly support the role of RIC8A in positively regulating YAP signaling in HEK293A cells.

We then asked whether RIC8A is also a potent positive regulator of YAP signaling in EGFR-mutant NSCLC cells. Indeed, across a panel of EGFR-mutant NSCLC cell lines, loss of *RIC8A* resulted in an increase in YAP phosphorylation at Ser127 site, an indicator of YAP inactivation (*Figure 4D*). In line with this finding, loss of *RIC8A* significantly suppressed the expression of YAP target genes in all tested cell lines (*Figure 4E*). Together, these data demonstrate that RIC8A positively regulates YAP signaling in EGFR-mutant NSCLC cells. Moreover, overexpression of YAP-5SA in representative HCC827 cell line blocked the effect of *RIC8A* loss on YAP signaling (*Figure 4F and G*). Importantly, YAP activation itself conferred resistance to EGFR inhibition and blocked *RIC8A* loss-induced synthetic lethality with EGFR inhibition (*Figure 4H–4J*). Collectively, these results suggest that loss of *RIC8A* synergizes with EGFR inhibition by attenuating YAP signaling in lung cancer.

To dive deeper into the connection between RIC8A and YAP signaling, we speculated that RIC8A positively regulates YAP signaling via Gα-Rho/Rac axis. We first tested whether RHOA inhibition could confer synthetic lethality with EGFR inhibition. Indeed, loss of *RHOA* increased YAP phosphorylation at Ser127 site (*Figure 4—figure supplement 2A*), decreased YAP-target gene expression (*Figure 4—figure supplement 2B*), and induced synthetic lethality with EGFR inhibition in HCC827 cells (*Figure 4—figure supplement 2C*). Then we examined whether RHOA activity is reduced by *RIC8A* loss. Unfortunately, after many attempts, we observed little decrease in the active RHOA signal upon *RIC8A* loss using a RHOA G-LISA Activation Assay Kit (*Figure 4—figure supplement 2D–*

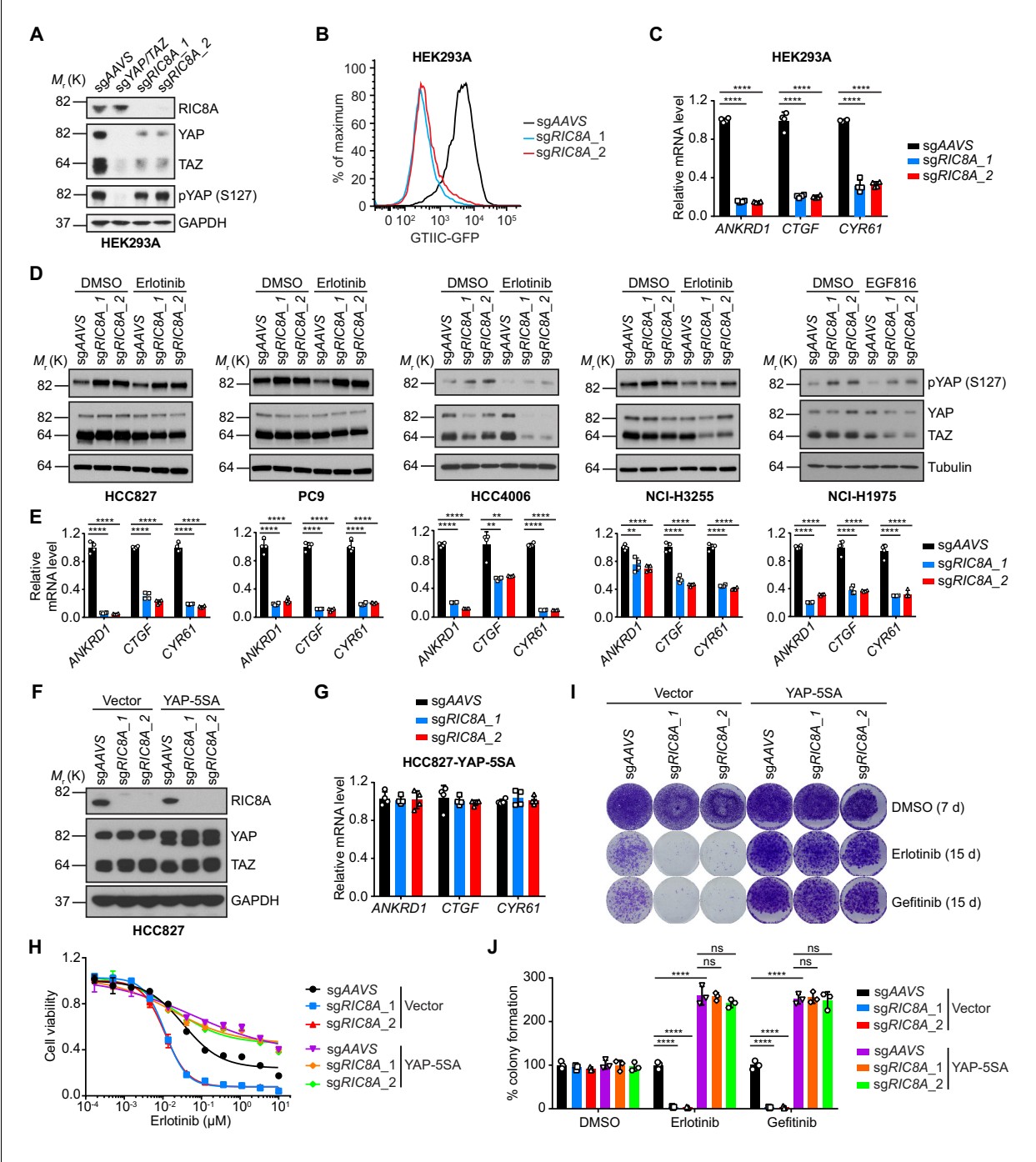

**Figure 4.** RIC8A loss attenuates YAP signaling to synergize with EGFR-TKI in EGFR-mutant NSCLC. (**A**) Immunoblots of indicated proteins in HEK293A cells upon knockout of indicated genes. GAPDH was used as a loading control. (**B**) RIC8A knockout decreases the YAP reporter activity in HEK293A cells, assessed by flow cytometry analysis of GFP. (**C**) Quantitative RT-PCR analysis of relative mRNA levels of YAP target genes in control or RIC8A knockout HEK293A cells. Error bars represent mean ± SD; n = 4. (**D**) Immunoblots of pYAP (S127) and YAP/TAZ in indicated EGFR-mutant NSCLC cell lines treated with DMSO or 1 μM EGFR-TKI (Erlotinib, except EGF816 for NCI-H1975) for 24 hr. Tubulin was used as a loading control. (**E**) Quantitative RT-PCR analysis of relative mRNA levels of YAP target genes in control or RIC8A knockout EGFR-mutant NSCLC cell lines. Error bars represent mean ± SD; n = 4. (**F**) Immunoblots of RIC8A and YAP/TAZ in indicated HCC827 cells to confirm RIC8A knockout and YAP-5SA overexpression. GAPDH was used as loading control. (**G**) Quantitative RT-PCR analysis of relative mRNA levels of YAP target genes in control or RIC8A knockout HCC827 cells with ectopic expression of constitutively active YAP (YAP-5SA). Error bars represent mean ± SD; n = 4. (**H**) Cell viability assessment by CellTiter-Glo assay of indicated HCC827 cell lines treated with serial dilutions of erlotinib for 72 hr. Error bars represent mean ± SD; n = 4. (**I**) Crystal violet staining colony formation assay of indicated HCC827 cell lines treated with DMSO, erlotinib (1 μM), or gefitinib (1 μM). (**J**) Quantification of colony formation in

*Figure 4 continued on next page*

Figure 4 continued

(I), shown as percentage of the Vector-sgAAVS sample. Error bars represent mean ± SD; n = 3. Statistical significance was tested using unpaired two-tailed t test (C and E) or ordinary two-way ANOVA (J); *p<0.05, **p<0.01, ***p<0.001, ****p<0.0001; ns, not significant.

The online version of this article includes the following source data and figure supplement(s) for figure 4:

**Source data 1.** Raw data from *Figure 4*.

**Figure supplement 1.** Overexpression of constitutively active YAP (YAP-5SA) blocks RIC8A loss induced reduction in YAP signaling and growth defect in HEK293A cells.

**Figure supplement 2.** RHOA signaling is involved in RIC8A-mediated regulation of EGFR TKI sensitivity.

*E*). However, this could be due to the fast dynamics of RHOA activation-inactivation cycle, making it difficult to capture the real-time changes in RHOA activity upon *RIC8A* loss. In addition, *ARHGAP29*, encoding a Rho GTPase activating protein, was previously reported to be a YAP target gene (*Qiao et al., 2017*). We observed that *RIC8A* loss caused significant decrease in the expression of *ARHGAP29* (*Figure 4—figure supplement 2F*), which could provide a negative-feedback mechanism to alleviate the decrease of RHOA activity resulting from *RIC8A* loss. Therefore, we believe our inability of detecting RHOA activity changes is most likely due to both technical reasons and the negative-feedback mechanism. Consistently, we observed a morphological alteration in HCC827 cells upon *RIC8A* loss (*Figure 4—figure supplement 2G*) and the decrease of Cofilin phosphorylation that is downstream of the RHOA-ROCK signaling (*Figure 4—figure supplement 2H*), indicating that loss of *RIC8A* indeed negatively impacted the output from RHOA activation. Moreover, treatment with Y-27632, the inhibitor of Rho-associated kinase ROCK, also induced synthetic lethality with EGFR inhibition in HCC827 cells (*Figure 4—figure supplement 2I–J*). Taken together, these data suggested the RIC8A-Gα-RHOA-YAP signaling axis is involved in the regulation of EGFR TKI sensitivity in EGFR-mutant NSCLC cells. Admittedly, RIC8A might regulate YAP signaling through other effectors, and comprehensive understanding of the signaling between RIC8A and YAP warrants future characterizations.

## *ARIH2* loss confers resistance to EGFR TKI in vitro and in vivo

To further validate the possible role of ARIH2 in EGFR TKI resistance, we introduced into HCC827 cells two independent sgRNAs targeting *ARIH2* and confirmed the *ARIH2* knockout efficiency by western blot analysis (*Figure 5A*). Loss of *ARIH2* in HCC827 cells decreased sensitivity to erlotinib (*Figure 5B*) and reduced apoptosis induction upon erlotinib treatment (*Figure 5C*). Accumulating evidence suggests that small subpopulations of cancer cells can survive strong EGFR inhibition by entering a DTP state, which could allow the emergence of heterogeneous EGFR TKI resistance mechanisms in EGFR-mutant lung cancers (*Sharma et al., 2010*; *Ramirez et al., 2016*). To examine whether loss of *ARIH2* could increase the reservoir of DTP cells, we treated control (sg*AAVS*) or *ARIH2*-deficient (sg*ARIH2*) HCC827 cells with 1 µM of erlotinib or gefitinib for three rounds with each treatment lasting for 72 hr. Our results demonstrated that loss of *ARIH2* facilitated more cells to enter into the DTP state (*Figure 5D–E*).

To test long-term effects of *ARIH2* loss, we first performed colony formation assays and confirmed that *ARIH2* loss substantially enhanced clonogenic cell survival upon EGFR inhibition (*Figure 5F–G*), which could be rescued by overexpression of CRISPR/Cas9-resistant ARIH2 (*Figure 5—figure supplement 1*). Next, we carried out a cell competition assay, in which unlabeled parental HCC827 cells were mixed in a ratio of 100:1 with RFP-labeled control or *ARIH2*-deficient cells and maintained in culture in the absence or presence of EGFR TKIs for three weeks (*Figure 5H*). Analysis of RFP-positive cells showed substantial outgrowth of the *ARIH2*-deficient cells under the pressure of EGFR inhibition (*Figure 5I–J*). The percentage of RFP-positive cells remained similarly in the absence of EGFR TKI (*Figure 5I–J*), together with no differential colony formation between control and *ARIH2*-deficient cells in the absence of EGFR TKI (*Figure 5F–G*), suggesting that loss of *ARIH2* has little or no effect on the basal proliferation of HCC827 cells. Likewise, drug resistance phenotype associated with *ARIH2* loss was also observed in NCI-H3255 cell line (*Figure 5—figure supplement 2A–F*). On the contrast, *ARIH2* loss had no effect on the EGFR TKI sensitivity in EGFR-WT NSCLC cells or normal cells (*Figure 5—figure supplement 3A–P*). Thus, our data

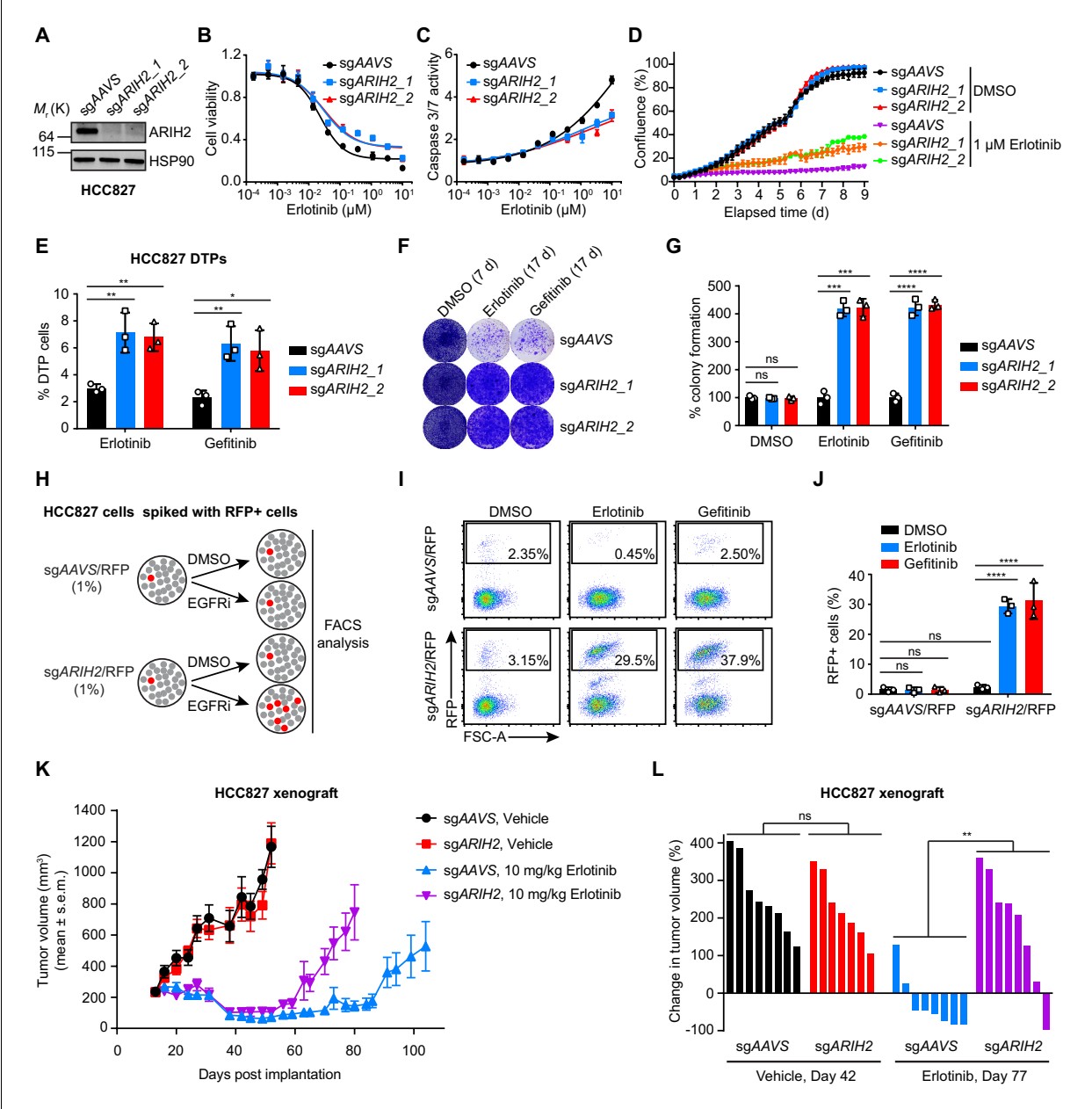

**Figure 5.** ARIH2 loss confers resistance to EGFR-TKI in vitro and in vivo. (**A**) Immunoblot of ARIH2 confirms the ARIH2 knockout efficiency in HCC827 cells. HSP90 was used as a loading control. (**B**) Cell viability assessment by CellTiter-Glo assay of control or ARIH2 knockout HCC827 cells treated with serial dilutions of erlotinib for 72 hr. Error bars represent mean ± SD; n = 4. (**C**) Activated caspase 3/7 measurement of control or ARIH2 knockout HCC827 cells treated with serial dilutions of erlotinib for 24 hr. Error bars represent mean ± SD; n = 4. (**D**) Kinetic cell proliferation assay monitored by IncuCyte for indicated HC827 cell lines cultured in the presence of DMSO control or 1 μM erlotinib over a 9 day period. (**E**) Drug-tolerant persister (DTP) cells were generated by treating control and ARIH2 knockout HCC827 cells with 1 μM of erlotnib or gefitinib for 9 d. Percentage of DTP cells is shown relative to DMSO-treated cells. Error bars represent mean ± SD; n = 3. (**F**) Crystal violet staining colony formation assay of control or ARIH2 knockout HCC827 cell lines treated with DMSO, erlotinib (1 μM), or gefitinib (1 μM). (**G**) Quantification of colony formation in (**F**), shown as percentage of the sg*AAVS* sample. Error bars represent mean ± SD; n = 3. (**H**) Schematic outline of the competitive proliferation assay to assess the selective outgrowth of ARIH2 knockout HCC827 cells upon EGFR-TKI treatment. RFP-negative HCC827 cells were spiked with approximately 1% RFP-positive sgRNA-infected HCC827 cells, control (sg*AAVS*) or ARIH2 knockout (sg*ARIH2*), and grown for 3 weeks in the absence or presence of 1 μM erlotinib or gefitinib. Cells were collected and analyzed for RFP positivity by FACS. (**I**) Selective outgrowth of ARIH2 knockout cells in the presence of EGFR-TKI in HCC827 cell line. The percentage of RFP-positive cells is indicated. FSC, forward scatter. (**J**) Quantification of the selective outgrowth of ARIH2 knockout cells in the presence of EGFR-TKI as shown in (**I**). Mean (three biological replicates) ± SD is shown. (**K**) ARIH2 knockout promotes acquired resistance to erlotinib in HCC827 xenograft model. Mice bearing HCC827 xenografts, control (sg*AAVS*) or ARIH2 knockout (sg*ARIH2*), were dosed once

*Figure 5 continued on next page*

*Figure 5 continued*

daily with 10 mg/kg erlotinib or vehicle for the indicated time frame. Data are represented as mean tumor volume (mm$^3$) ± s.e.m., n = 8 mice for each line. (**L**) Percentage change in tumor volume compared to baseline (the start of dosing, day 13 post-implantation) for individual cell xenografts treated for 29 d (day 42 post-implantation) with vehicle or 64 d (day 77 post-implantation) with 10 mg/kg erlotinib. Statistical significance was tested using unpaired two-tailed t test (**E, G and L**) or ordinary two-way ANOVA (**J**); *p<0.05, **p<0.01, ***p<0.001, ****p<0.0001; ns, not significant.

The online version of this article includes the following source data and figure supplement(s) for figure 5:

**Source data 1.** Raw data from *Figure 5*.
**Figure supplement 1.** Crystal violet staining colony formation assay showing the rescued phenotype by overexpression of CRISPR/Cas9-resistant ARIH2.
**Figure supplement 2.** ARIH2 loss confers resistance to EGFR-TKI.
**Figure supplement 3.** ARIH2 loss exhibits no effect on EGFR TKI sensitivity in EGFR-WT NSCLC cells or normal human bronchial epithelial cells.
**Figure supplement 4.** Graph showing the gain in body weight relative to day 13 post-implantation (the start of dosing).

provide a chain of evidence to demonstrate that *ARIH2* loss reduces the sensitivity of EGFR-mutant NSCLC cells to EGFR inhibition and promotes acquired resistance.

We continued to assess the resistant effect of *ARIH2* loss to erlotinib treatment in vivo. We established xenografts of control or *ARIH2*-deficient HCC827 cells in nude mice. Mice were enrolled in the study once tumors had reached approximately 200 mm$^3$ in size, and were randomly assigned to receive either vehicle or 10 mg/kg erlotinib once daily for the duration of the study (*Figure 5K–L* and *Figure 5—figure supplement 4*). As shown in *Figure 5K*, loss of *ARIH2* alone had little effect on tumor growth in the vehicle treatment group. Erlotinib treatment efficiently suppressed tumor growth (compared to vehicle treatment), with resistance emerging after approximately 90 days of continuous erlotinib treatment in control xenograft tumors (*Figure 5K*). Importantly, loss of *ARIH2* significantly accelerated the development of resistant tumors (*Figure 5K–L*). Taken together, these data further support the notion that loss of *ARIH2* confers resistance to EGFR inhibition in EGFR-mutant NSCLC.

## Mechanistic insights into *AIRH2* loss-mediated EGFR TKI resistance

To gain mechanistic insights into how *ARIH2* loss confers resistance to EGFR inhibition, we compared the global protein changes between control and *ARIH2*-deficient HCC827 cells by quantitative mass spectrometry analysis (*Figure 6A*). Using a stringent criteria (|Log2 fold change| $\geq$ 0.9 and $P$ value $\leq$ 0.001), we observed that 46 proteins significantly increased and 13 proteins decreased upon *ARIH2* loss (*Figure 6A*). We surmised that loss of *ARIH2* might increase the abundance of certain essential proteins to survive EGFR inhibition. Therefore, we searched the 46 proteins with increased abundance upon *ARIH2* loss and were particularly interested in a few hits for further characterization, including METAP2, ALDOA and PSAT1. METAP2, also known as methionine aminopeptidase 2, is a eukaryotic initiation factor 2 (eIF2)-associated glycoprotein which possesses dual functions of regulating global protein synthesis rate and co-translationally removing the N-terminal methionine from nascent proteins (*Datta, 2000*). ALDOA (fructose-bisphosphate aldolase A) and PSAT1 (phosphoserine aminotransferase) are important enzymes involved in glycolysis and serine biosynthesis pathways, respectively. We first validated by immunoblotting analysis that loss of *ARIH2* indeed increased protein levels of METAP2, ALDOA and PSAT1 in HCC827 cells (*Figure 6B*), without affecting their corresponding mRNA levels (*Figure 6C*), suggesting a possible post-transcriptional regulation mechanism. Loss of *ARIH2* also led to increase of METAP2 and ALDOA proteins, but not PSAT1 protein, in NCI-H3255 cells (*Figure 6—figure supplement 1A–B*). Moreover, loss of *ARIH2* or *CUL5* also dramatically increased protein abundance of METAP2, ALDOA and PSAT1 in HEK293T cells (*Figure 6—figure supplement 1C*), suggesting a general mechanism of regulation of these proteins by ARIH2 and CUL5 complex.

We continued to examine whether increased abundance of these proteins could confer resistance to EGFR inhibition in EGFR-mutant NSCLC cells. We first focused on METAP2, the abundance of which increased most upon *ARIH2* loss, and ectopically overexpressed a hemagglutinin (HA) tagged METAP2 (HA-METAP2) in HCC827 cells (*Figure 6D*). A long-term colony formation assay of the vector- and HA-METAP2-expressing HCC827 cells demonstrated that METAP2 overexpression confers resistance to EGFR inhibition (*Figure 6E–F*). Given its role in regulating global protein synthesis, we sought to assess de novo protein synthesis in both vector- and METAP2-overexpressing cells by

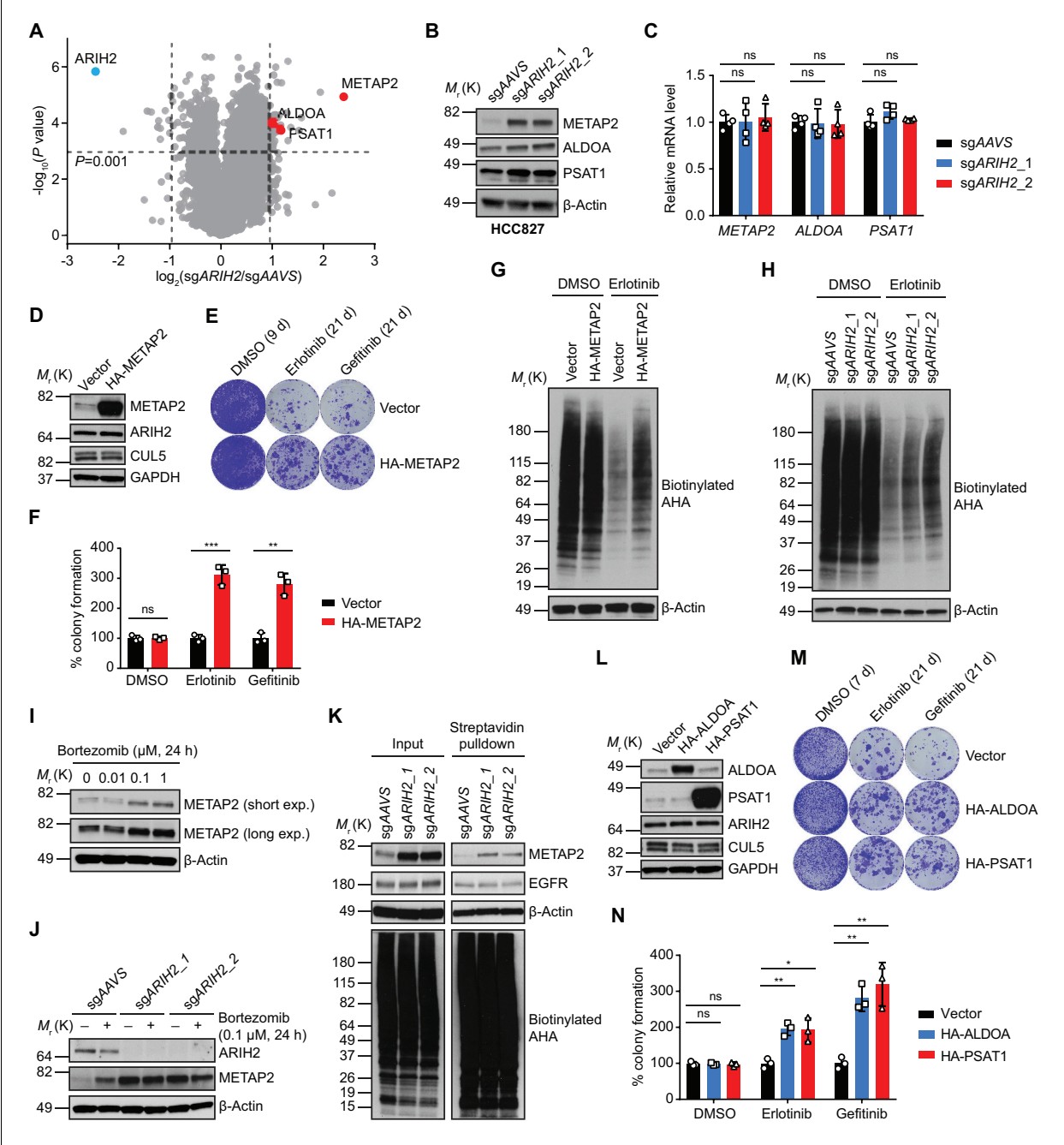

**Figure 6.** Mechanistic insights into ARIH2 loss-mediated EGFR-TKI resistance. (A) Mass spectrometry analysis of global protein changes between control and ARIH2-deficient HCC827 cells. (B) Immunoblots of indicated proteins in control and ARIH2-deficient HCC827 cells. β-Actin was used as a loading control. (C) Quantitative RT-PCR analysis of relative mRNA levels of indicated genes in control or ARIH2-deficient HCC827 cells. Error bars represent mean ± SD; n = 4. (D) Immunoblots of indicated proteins showing ectopic expression of HA-METAP2 in HCC827 cells. GAPDH was used as a loading control. (E) Crystal violet staining colony formation assay of HCC827-Vector or HCC827-HA-METAP2 cell lines treated with DMSO, erlotinib (1 µM), or gefitinib (1 µM). (F) Quantification of colony formation in (E), shown as percentage of the HCC827-Vector sample. Mean (three biological replicates) ± SD is shown. (G) De novo protein synthesis of HCC827-Vector or HCC827-HA-METAP2 cells after treatment with DMSO or erlotinib (1 µM, 24 hr) as determined by L-azidohomoalanine (AHA) labeling. Cells were starved of methionine for 1 hr and incubated with AHA for 1 hr. Lysates were subjected to a Click-iT chemistry reaction to switch azido-modified nascent proteins to alkyne-biotin, and visualized by Streptavidin-HRP immunoblotting. β-Actin was used as a loading control. (H) De novo protein synthesis of control or ARIH2-deficient HCC827 cells after treatment with DMSO or erlotinib (1 µM, 24 hr) as determined by AHA labeling. β-Actin was used as a loading control. (I) Immunoblot of METAP2 in HCC827 cells upon proteasome inhibitor bortezomib treatment. β-Actin was used as a loading control. (J) Immunoblots of ARIH2 and METAP2 in control or ARIH2 knockout HCC827 cells upon bortezomib treatment. β-Actin was used as a loading control. (K) De novo METAP2 protein synthesis in control or ARIH2-

*Figure 6 continued on next page*

*Figure 6 continued*

deficient HCC827 cells as determined by AHA labeling and streptavidin pulldown. β-Actin was used as a loading control. (**L**) Immunoblots of indicated proteins showing ectopic expression of HA-ALDOA and HA-PSAT1 in HCC827 cells. GAPDH was used as a loading control. (**M**) Crystal violet staining colony formation assay of HCC827-Vector, HCC827-HA-ALDOA or HCC827-HA-PSAT1 cells treated with DMSO, erlotinib (1 μM), or gefitinib (1 μM). (**N**) Quantification of colony formation in (**M**), shown as percentage of the HCC827-Vector sample. Mean (three biological replicates) ± SD is shown. Statistical significance was tested using unpaired two-tailed t test (**C, F** and **N**); *$p<0.05$, **$p<0.01$, ***$p<0.001$, ****$p<0.0001$; ns, not significant.

The online version of this article includes the following source data and figure supplement(s) for figure 6:

**Source data 1.** Raw data from *Figure 6*.
**Figure supplement 1.** ARIH2 loss increases protein abundance of METAP2, ALDOA and PSAT1.

L-azido-homoalanine (AHA) labeling (*Figure 6G*). EGFR inhibition by erlotinib treatment drastically inhibited nascent protein synthesis (*Figure 6G*), in line with its effect on overall growth inhibition. Importantly, cells with METAP2 overexpression exhibited increased protein synthesis, compared to vehicle-expressing cells, when challenged by erlotinib treatment (*Figure 6G*). Furthermore, loss of *ARIH2*, which is able to increase METAP2 protein level, also increased nascent protein synthesis upon EGFR inhibition (*Figure 6H*). Together, these data suggest that *ARIH2* loss confers resistance to EGFR inhibition, at least in part, by maintaining higher protein synthesis rate through METAP2.

Next, we investigated how ARIH2 regulates METAP2 protein level. EGFR inhibition did not alter METAP2 protein level in both control and *ARIH2*-deficient HCC827 cells (*Figure 6—figure supplement 1D–F*), ruling out the possibility that METAP2 is a downstream effector of EGFR which could be regulated by ARIH2. Proteasome inhibitor (Bortezomib) treatment increased METAP2 protein abundance in HCC827 cells (*Figure 6I*), suggesting that METAP2 protein level is regulated by the proteasome-dependent degradation pathway. Moreover, bortezomib induced METAP2 protein level increase was only observed in control cells but not in *ARIH2*-deficient cells (*Figure 6J*), indicating that proteasome-mediated METAP2 protein degradation is dependent on ARIH2. We further attempted to examine METAP2 ubiquitination in HCC827 cells. Unfortunately, we were unable to detect endogenous METAP2 protein ubiquitination (*Figure 6—figure supplement 1G*), likely due to that the METAP2 ubiquitination level is too low to be detected or ARIH2 regulates METAP2 degradation indirectly. Next, we also wondered whether ARIH2 regulates METAP2 protein translation/synthesis. Specifically, we assessed de novo METAP2 protein synthesis in control or *ARIH2*-deficient HCC827 cells by AHA labeling followed by streptavidin pulldown and showed that loss of *ARIH2* increased nascent METAP2 protein synthesis (*Figure 6K*), suggesting that ARIH2 indeed regulates METAP2 protein translation. As a control, EGFR protein synthesis remained unchanged upon *ARIH2* loss (*Figure 6K*). Taken together, these data demonstrate that ARIH2 is capable of regulating both protein translation and protein degradation of METAP2. Future studies are required to gain a comprehensive picture of the regulation of METAP2 protein level by ARIH2.

Lastly, we ectopically overexpressed HA-ALDOA or HA-PSAT1 in HCC827 cells as well (*Figure 6L*). Long-term colony formation assays showed that overexpression of ALDOA or PSAT1 is capable to promote resistance to EGFR inhibition (*Figure 6M–N*). Thus, we concluded that loss of ARIH2 confers resistance to EGFR inhibition in EGFR-mutant NSCLC cells by integrating multiple mechanisms.

## Discussion

Understanding and overcoming resistance to EGFR TKIs remain a major challenge in NSCLC. Several studies have previously employed loss- or gain-of-function screens to profile genetic interactions with mutant EGFR in NSCLC cells to look for synthetic lethality (*Bivona et al., 2011*; *de Bruin et al., 2014*; *Sharifnia et al., 2014*; *Lantermann et al., 2015*; *Liao et al., 2017*). However, they utilized either small interfering RNA (siRNA) or short hairpin RNA (shRNA) approach or focused on small subsets of genes (such as tumor suppressors, oncogenes and genes encoding kinases), and identified distinct genetic modifiers of EGFR TKI sensitivity. Herein in this study, we presented an unbiased genome-scale CRISPR-Cas9 screening strategy to systematically capture the breadth of genetic modifiers of mutant EGFR dependency in NSCLC. We purposely selected HCC827, a cell line with high sensitivity to EGFR TKI, and generated resistance by applying clinically relevant concentration of erlotinib for a sustained period, allowing the identification of both negative and positive

regulators of EGFR TKI sensitivity simultaneously. Our genome-wide CRISPR screen successfully revealed a number of known causative genes and signaling pathways associated with EGFR TKI resistance, such as YAP signaling, PI3K-mTOR signaling and RAS-MAPK signaling pathways, reinforcing their significance in mediating mutant EGFR dependency in NSCLC. More importantly, we uncovered a list of previously unrecognized genes whose deletion caused synthetic lethality with or conferred resistance to EGFR TKI treatment, broadening our understanding of EGFR signaling regulation. Importantly, we determined the copy number of these genes by searching the database (https://cansar.icr.ac.uk/cansar/cell-lines/HCC-827/copy_number_variation/no%20signal/) and did not find gain of copy numbers for any of the genes studied herein, ruling out the false positive possibility of artifacts from targeting genes in amplicons.

Among those newly identified mutant EGFR dependencies, GPCR/G-protein signaling module stood out as a strong vulnerability with EGFR inhibition. Particularly, genetic ablation as well as pharmacological antagonism of the LPARs strongly sensitized EGFR-mutant NSCLC cells to EGFR inhibition. Interestingly, LPAR signaling has been shown to contribute to malignancy and chemotherapy resistance in various tumors (*Yung et al., 2014*; *Hashimoto et al., 2016*). Molecularly, GPCRs initiate downstream signaling cascades through activating heterotrimeric G proteins consisting of Gα subunit (G12/13, Gq/11, Gi/o, or Gs) and Gβγ heterodimer in concert with other effector proteins (*Dorsam and Gutkind, 2007*). Importantly, inactivation of GPCR related effectors, such as RIC8A, PKN2, GNB2 (encoding Gβ2 subunit) or GNA13 (encoding Gα13 subunit), strongly synergized with EGFR inhibition in EGFR-mutant NSCLC cells, revealed by our CRISPR screen. Furthermore, loss of *KCTD5*, a previously reported negative regulator of Gβ protein stability (*Brockmann et al., 2017*), promoted resistance to EGFR inhibition. Together, these findings suggest that EGFR-mutant NSCLC depends on parallel GPCR signaling to modulate mutant EGFR addiction. *RIC8A*, the most prominent synergy hit from our screen, possesses dual functions as a molecular chaperone required for nascent Gαq/i/13 protein folding and initial membrane association as well as a guanine nucleotide exchange factor (GEF) for Gαq/i/13 (*Nishimura et al., 2006*; *Gabay et al., 2011*; *Chan et al., 2013*; *Kant et al., 2016*). Interestingly, LPAR signaling coupled with G12/13 has been shown to activate YAP pathway in HEK293A cells (*Yu et al., 2012*). Moreover, majority of human uveal melanomas, driven by activating mutations in Gαq/11 proteins, depend on YAP signaling for tumorigenesis (*Yu et al., 2014*). *Ric-8A* gene deletion significantly suppressed tumorigenesis in a mouse model of oncogenic Gαq-driven melanoma (*Patel and Tall, 2016*), implying its regulation of YAP signaling pathway. Here, we extended the regulation of YAP pathway by GPCR and G proteins to RIC8A, and demonstrated, for the first time, that RIC8A positively regulates YAP signaling to modulate EGFR TKI sensitivity in EGFR-mutant NSCLC cells. Therefore, targeting RIC8A might be promising to prevent EGFR TKI resistance in lung cancer. YAP signaling pathway has also been shown to mediate resistance to other targeted therapies, such as RAF and MEK inhibition (*Lin et al., 2015*). Whether targeting RIC8A could inhibit YAP signaling and consequently enhance treatment response in those circumstances is of great interest and awaits future investigation.

On the other hand, reduced NF1 (neurofibromin) expression (*de Bruin et al., 2014*) or loss of PTEN (*Sos et al., 2009*) has been observed in clinical specimens with acquired resistance to EGFR TKI treatment. Both of them strongly scored as resistance hits in our CRISPR screen. Among other positively selected hits, *ARIH2*, *CUL5*, *RNF7* and *UBE2F*, falling into an ubiquitination related functional module, significantly stood out. Cullin-RING ligases (CRLs), comprising the largest subfamily of E3 ubiquitin ligases, are activated by ligation of the ubiquitin-like protein NEDD8 to a conserved cullin lysine (*Lamsoul et al., 2016*). The NEDD8-conjugating enzyme UBE2F, via specific interaction with E3 ligase RNF7 (also known as RBX2), specifically neddylates and consequently activates cullin-5 (CUL5), but not other cullin proteins (*Huang et al., 2009*). ARIH2 (also known as TRIAD1), a member of the RING-in-between-RING (RBR) E3 ubiquitin ligase family (*Dove and Klevit, 2017*), binds to and is activated specifically by activated CRL5 complex (*Kelsall et al., 2013*). ARIH2 is essential for embryogenesis and has been shown to modulate inflammatory responses (*Lin et al., 2013*; *Kawashima et al., 2017*). The association of CRL5 complex with cancer biology has started to emerge. For instance, *CUL5* deficiency has been shown to promote SCLC metastasis by stabilizing integrin β1 (*Zhao et al., 2019*). Additionally, CRL5 complex also plays important roles in modulating multiple aspects of the cellular response to heat shock protein 90 (HSP90) inhibition (*Samant et al., 2014*). Here, for the first time, we emphasized the importance of ARIH2-CRL5 complex in mediating EGFR TKI resistance in NSCLC. Through a proteomics study, we identified METAP2 (methionine

aminopeptidase 2), a protein that has been shown to be involved in NSCLC (*Shimizu et al., 2016*), the abundance of which was dramatically increased upon *ARIH2* loss. We further demonstrated that increased METAP2 level was, at least in part, responsible for the drug resistance phenotype resulting from *ARIH2* loss. However, identification of direct substrates of ARIH2-CRL5 complex as well as elucidating their involvement in EGFR TKI resistance remains to be studied more systematically in the future.

In summary, while clinical studies are necessary to confirm these newly revealed dependencies of mutant EGFR in NSCLC, our genome-wide CRISPR-Cas9 genetic screen together with validation and mechanistic studies expanded the understanding of the heterogeneity of EGFR TKI responses in lung cancer.

# Materials and methods

## Cell culture

Human cancer cell lines were originated from the CCLE (*Barretina et al., 2012*), banked at Novartis Cell Bank, authenticated by single-nucleotide polymorphism analysis and routinely tested as mycoplasma-free. All cell lines were maintained at 37°C with 5% $CO_2$. HEK293T, HEK293A and A549 cells were cultured in DMEM medium (Gibco #11995–040) supplemented with 10% fetal bovine serum (FBS) (Seradigm #1500–500) and penicillin (100 units/ml)-streptomycin (100 μg/ml) (Gibco #15140–122). HCC827, HCC4006, NCI-H3255, PC9, NCI-H1975, NCI-H1299 and NCI-H460 cells were cultured in RPMI medium 1640 (Gibco #22400–071) supplemented with 10% FBS and penicillin-streptomycin. BEAS-2B cells were cultured in BEGM bronchial epithelial cell growth medium bulletkit (Lonza #CC-3170).

## Compounds

Erlotinib (S1023), Gefitinib (S1025) and Bortezomib (PS-341) were obtained from Selleckchem. NVP-EGF816 and LPAR antagonists (Compound two and AM095) were synthesized at Novartis Institutes for Biomedical Research. Ki16425 (Cat #5056) was purchased from Tocris. Y-27632 (Cat# Y0503) was purchased from Sigma-Aldrich. Drugs for in vitro studies were dissolved in DMSO (ATCC 4-X) to yield 10 mM stock solutions and stored at −20°C.

## Plasmid constructs

For CRISPR-Cas9 mediated gene ablation, the pNGx-LV-c004-3xFlag-Cas9 construct and pNGx-LV-g003 lentiviral backbone for sgRNA cloning were described previously (*DeJesus et al., 2016*). For each sgRNA expression clone, spacer-encoding sense and anti-sense oligonucleotides with appropriate overhangs were synthesized (IDT), annealed, cloned into the BbsI restriction site of the pNGx-LV-g003 backbone. Insertion was verified by DNA sequencing.

For cDNA overexpression, pLVX-EF1α-IRES-Puro vector was purchased from TaKaRa (cat# 631988). Amino-terminally HA-tagged cDNAs were amplified by PCR using Q5 High-Fidelity 2X Master Mix (NEB Inc, #M0492S) with the following thermocycling conditions: 30 s at 98°C, 30 cycles of 10 s at 98°C, 20 s at 55°C and then 90 s at 72°C, followed by 2 min at 72°C and holding at 4°C. PCR products were purified with QIAquick PCR Purification Kit (Qiagen #28104) following manufacturer's instructions. Purified PCR products and empty pLVX-EF1α-IRES-Puro vector were digested by EcoRI-HF (NEB Inc, #R3101S) and XbaI (NEB Inc, #R0145S) in 1X CutSmart Buffer (NEB Inc, #B7204S) at 37°C for overnight. Digested fragments were run in agarose gel and purified with QIAquick Gel Extraction Kit (Qiagen #28704) following manufacturer's instructions. Ligation was performed (3:1, insert: vector molar ratio) with T4 DNA Ligase (NEB Inc, #M0202) in 1X T4 DNA Ligase Reaction Buffer (NEB Inc, #B0202S) at room temperature for 4 hr. For bacterial transformation, One Shot Stbl3 Chemically Competent *E. coli* (Thermo Fisher Scientific, #C737303) was used according to the manufacturer's instructions. Plasmid isolation was performed using QIAprep Spin Miniprep Kit (Qiagen #27104) following manufacturer's instructions. Insertion was verified by DNA sequencing.

## sgRNA target sequences

*AAVS*_g1: 5'−3' GTTAATGTGGCTCTGGTTCT;
*GNB2*_g1: 5'−3' TCTTTGCCAGGTGCCCACGG;

*LPAR2*_g1: 5'−3' GCCCGCGAAGAGGTCAGCCG;
*PKN2*_g1: 5'−3' TCTGCAAATAAAGTACCCTG;
*RIC8A*_g1: 5'−3' GGAGTGCCGTTAGCAGGAAG;
*RIC8A*_g2: 5'−3' GGAGCCGCAAGTCAAAGAAC;
*USP22*_g1: 5'−3' GCATATTCACGAGCATGCGA;
*YAP1*_g1: 5'−3' ACATCGATCAGACAACAACA;
*ARIH2*_g1: 5'−3' ATATCTCTGAAACTTGCCAG;
*ARIH2*_g2: 5'−3' AGTGCTGCTCCCAGCAGCTG;
*CUL5*_g1: 5'−3' AGCTTGTTTACATAATCCGC;
*RNF7*_g1: 5'−3' CCTCAAGAAGTGGAACGCGG;
*KCTD5*_g1: 5'−3' AAGTGGGTCCGACTCAACGT;
*PDCD10*_g1: 5'−3' CAACTCACCTCATTAAACAC;
*BTAF1*_g1: 5'−3' GTGAAGTGGATCCTAAAGAG;

## Lentivirus production

For single guide RNA (sgRNA) lentivirus production, 1 µg of sgRNA construct was co-transfected into HEK293T cells with approximately 80% confluence in a well of the 6-well tissue culture plate (Corning #353046) along with 1 µg packaging (Δ8.9) and 0.25 µg envelope (VSV-G) expression plasmids using 6.75 µl FuGENE 6 Transfection Reagent (Promega #E2692) according to the manufacturer's instructions. Cell culture medium was replaced at 16 hr after transfection, and lentivirus-containing supernatant was harvested at 48 hr and 72 hr post-transfection. Viral supernatant was filtered through a 0.45 µm cellulose acetate filter (Thermo Fisher Scientific #723–9945), aliquoted and stored at −80°C for later use.

For cDNA ectopic expression lentivirus production, 6 µg of Ready-to-Use Lentiviral Packaging Plasmid Mix (Cellecta #CPCP-K2A) and 4.8 µg of cDNA expression construct were co-transfected into HEK293T cells with approximately 80% confluence in a 100 mm cell culture dish (Corning #430167) using 32.4 µl FuGENE 6 Transfection Reagent (Promega #E2692) according to the manufacturer's instructions. Cell culture medium was replaced at 16 hr after transfection, and lentivirus-containing supernatant was harvested at 48 hr and 72 hr post-transfection. Viral supernatant was filtered through a 0.45 µm cellulose acetate filter, mixed with 1/3 vol of Lenti-X Concentrator (Clontech #631232) and incubated at 4°C for overnight. Viral particles were pelleted by centrifugation at 1,500 g for 45 min at 4°C. The pellet was then gently resuspended with 1 ml of complete DMEM cell culture medium, appropriately aliquoted and stored at −80°C for later use.

## Generation of cell lines

Human cancer cell lines with constitutive Cas9 expression were generated by lentiviral infection and antibiotic selection. Cas9 expression was confirmed by immunoblotting and gene editing efficiency was tested as follows. Cas9-expressing cells were infected at a low (~0.5) multiplicity of infection (MOI) with lentivirus expressing either a control *AAVS* sgRNA or sgRNA targeting essential genes *PSMD1* and *PSMA3* and then selected with puromycin. Cells were subsequently seeded in 6-well tissue culture plates, cell culture medium was exchanged 3 days later and the experiment was terminated at day 7. Cells were trypsinized, resuspended in cell culture medium and the live cell count was determined by trypan blue exclusion on a ViCELL instrument (Beckman Coulter).

To generate YAP reporter cell line, HEK293A cells were infected by lentiviruses expressing Cas9 and GTIIC-GFP reporter, and clonal cells were selected and experimentally validated. To knockout specific genes, Cas9-expressing cells in 6-well tissue culture plate were infected by lentivirus expressing sgRNA targeting gene of interest in the presence of complete cell culture medium supplemented with 8 µg/ml polybrene (AmericanBio #AB01643-00001). Following infection for 18 hr, cell culture medium was replaced by complete cell culture medium. Following another 24 hr, cell culture medium was replaced by complete cell culture medium containing 1 µg/ml puromycin (Mediatech #MT-61–385-RA) and mutant cell pools stably expressing the sgRNA were selected. Stable cell lines with ectopic cDNA overexpression were generated in the same manner. Gene silencing efficiency or cDNA overexpression was determined by immunoblotting assay.

## Genome-wide sgRNA library design and construction

The genome-wide sgRNA library targeting 18,360 protein-coding genes (~5 sgRNAs per gene) was adapted from published sequences (*Sanjana et al., 2014*). For genes lacking published sgRNA sequence information, new sgRNAs were designed for these targets that contained an NGG PAM motif, filtering for GC content >40% and<80%, eliminating homopolymer stretches > 4, and removing any guides with off-target locations having <4 mismatches across the genome. The sgRNA library was constructed using chip-based oligonucleotide synthesis to generate spacer-tracrRNA-encoding fragments that were PCR-amplified and cloned as a pool into the BpiI site of the pRSI16 lentiviral plasmid (Cellecta). Sequencing of the plasmid pool showed robust normalization, with >90% clones present at a representation of ±5 fold from the median counts in the pool.

Library packaging was performed as described previously (*Zeng et al., 2018*). The sgRNA libraries were packaged into lentiviral particles using HEK293T cells. Packaging was scaled up by growing cells in cell stacks (Corning). For each cell stack, 210 million cells were transfected 24 hr after plating using 510.3 µl TransIT Transfection Reagent (Mirus Bio, #MIR2700) diluted in 18.4 ml Opti-MEM (Gibco #11058021) that was combined with 75.6 µg of the sgRNA library and 94.5 µg of Ready-to-Use Lentiviral Packaging Plasmid Mix (Cellecta #CPCP-K2A). The next day, the transfected cells received fresh medium. 72 hr post-transfection, lentivirus was collected, filtered, aliquoted, and frozen at −80°C. Viral titer was determined using the Lenti-X qRT-PCR Titration Kit (Clontech #631235) and was typically in the range of $5 \times 10^6$ transforming units/ml.

For genome-wide screens, sgRNA libraries were transduced at a multiplicity of infection (MOI) of 0.5, aiming for coverage of, on average, 1,000 cells per sgRNA reagent. MOI was determined by using a 12-point dose-response ranging from 0 to 400 µl of viral supernatants in the presence of 5 µg/ml polybrene and measuring infection rate by FACS as a percentage of red fluorescent protein (RFP)-positive cells. Selection was optimized by determining the puromycin dose required to achieve >95% cell killing in 72 hr. Cell viability was measured for a 6-point dose-response ranging from 0 to 20 µg/ml puromycin using the CellTiter-Glo assay (Promega).

## CRISPR-Cas9 screening for determinants of EGFR-TKI sensitivity

HCC827-Cas9 cells were seeded into cell stacks (Corning). 24 hr after plating, the culture medium was replaced with fresh medium containing 5 µg/ml polybrene and lentiviral sgRNA library at an MOI of 0.5. 24 hr after infection, the culture medium was replaced with fresh medium containing 2 µg/ml puromycin. 72 hr after puromycin selection, cells were trypsinized and an aliquot of cells was analyzed by FACS to confirm infection and selection efficiency, and the percentage of RFP-positive cells was >90%. 100 million cells, representing the baseline of sgRNA representation, were harvested and snap-frozen using liquid nitrogen. The remaining cells were plated into new cell stacks at 110 million cells per cell stack. The next day, cells were treated with DMSO or 1 µM erlotinib, respectively. Cell culture medium containing DMSO or 1 µM erlotinib was replenished every 3 days. Cells were maintained in culture and split as needed to ensure confluence did not exceed 90%. After 3 weeks of treatment, 100 million cells from each condition were harvested.

## Illumina library construction and sequencing

Genomic DNA was isolated using the QIAamp DNA Blood Maxi Kit (Qiagen #51194) as directed by manufacturer and quantified using PicoGreen (Invitrogen). Illumina sequencing libraries were generated using PCR amplification with primers specific to the genome integrated lentiviral vector backbone sequence. The resulting Illumina libraries were purified using 1.8x SPRI AMPure XL beads (Beckman Coulter) following the manufacturer's instructions and qPCR quantified using primers specific to the Illumina sequences using standard methods. Illumina sequencing libraries were pooled and sequenced with a HiSeq 2500 instrument (Illumina). The number of reads was adjusted to cover each sgRNA with approximately 1000 reads.

## CRISPR screen data analysis

Raw sequencing reads were aligned to the appropriate library using Bowtie (*Langmead et al., 2009*), allowing for no mismatches, and counts were generated. The R software package DESeq2 (*Love et al., 2014*) was used to evaluate differential sgRNA representation in the form of log2 fold change between the erlotinib-treated and DMSO-treated samples for EGFR-TKI screen. A robust

z-score was calculated using the median and mean-absolute deviation for the calculated fold changes across the entire sgRNA library. For gene-based hit calling, the sgRNAs were ranked by the robust z-score, and the statistical significances for each gene enriched toward higher rank (RSA up) and the lower rank (RSA down) were evaluated using the Redundant siRNA Activity (RSA) algorithm (König et al., 2007). The RSA score is a statistical score (log10 ($P$ value)) representing the probability of a gene hit based on the collective activities of multiple sgRNAs per gene. It is a measure of how significantly the rank order of sgRNAs against a given gene differs from the population of other sgRNAs in the library. Selected hits were searched against STRING database version 11.0 for mapping protein interaction network and Reactome database version 67 for pathway analysis.

## Short-term cell proliferation assay

Cells were seeded in 384-well microplates (Corning #3570) at a density of 1,000 cells in 30 µl of complete cell culture medium per well and allowed to adhere overnight. Cells were treated in quadruplicate with 6 µl of serial three-fold dilutions of compound in complete cell culture medium (final DMSO concentration = 0.1%). Following drug exposure for 72 hr, 25 µl of CellTiter-Glo reagent (Promega #G7572) per well was added and plates were incubated at room temperature for 20 min. Luminescence was read in an EnVision Multimode Plate Reader (PerkinElmer). Assay data were normalized to DMSO values and plotted using a four-parameter concentration-response model in GraphPad Prism 7. The figures show the mean ± standard deviation of quadruplicate values from representative experiments.

## Activated caspase 3/7 assay

Cells were seeded in 384-well microplates (Corning #3570) at a density of 2,000 cells per well and allowed to adhere overnight. Cells were treated in quadruplicate with serial three-fold dilutions of compound in complete cell culture medium (final DMSO concentration = 0.1%). Following drug exposure for 24 hr, caspase 3/7 activity was measured using the Caspase-Glo 3/7 Assay System (Promega #G8092) according to the supplier's instructions. Luminescence was read in an EnVision Multimode Plate Reader (PerkinElmer).

## Kinetic cell growth assay

Cells were seeded in 6-well tissue culture plates at a density of $1 \times 10^5$ cells per well and allowed to adhere overnight. Cells were then treated in triplicate with DMSO control or 1 µM erlotinib. Photomicrographs (36 images per well) were taken every 6 hr using an IncuCyte live cell imager (Essence BioSciences) and confluence of the cultures was measured using IncuCyte software (Essence BioSciences).

## Generation of drug-tolerant persister (DTP) cells

DTP cells were generated according to protocols described previously (Sharma et al., 2010). In brief, cells were treated with 1 µM of erlotinib or gefitinib for three rounds, with each treatment lasting 72 hr. Viable cells remaining attached on the dish at the end of the 9 d drug treatment were considered to be DTPs and were collected for analysis. The live cell count was determined by trypan blue exclusion on a ViCELL instrument (Beckman Coulter). Percentage of DTPs was calculated by comparing the number of DTPs to the number of cells at the end of the 9 d DMSO treatment.

## Long-term clonogenic growth assay/colony formation assay

Cells were seeded into 6-well tissue culture plates ($5 \times 10^4$ to $1 \times 10^5$ cells per well, depending on the growth rate) and allowed to adhere overnight in complete cell culture medium. The next day, medium was replaced by complete cell culture medium containing appropriate drugs or DMSO as vehicle control. Cells were exposed to vehicle for 7–10 days or indicated drug for 3–5 weeks, with medium change and fresh drug added twice a week. At the end of treatment, remaining cells were gently washed with PBS, fixed/stained with 0.2% crystal violet (Fisher Scientific #C581-25) in 4% paraformaldehyde and incubated at room temperature for 30 min. Cells were washed three times with water to remove excessive dye and allowed to air dry. Pictures of stained cells were taken using an EPSON Perfection V600 scanner. Colony formation was quantified using the ColonyArea ImageJ plugin which provides information about the intensity percentage taking into consideration not only

the area covered by the colonies, but also the intensity of staining as a direct relation to the number of cells in a colony (*Guzmán et al., 2014*).

## Competitive proliferation assay

For competitive proliferation assay, HCC827 parental cells (RFP negative) were mixed with HCC827-sg*AAVS*/RFP or HCC827-sg*ARIH2*/RFP cells (100:1) and cultured in the presence of DMSO control or EGFR-TKI (erlotinib or gefitinib; 1 µM) for 3 weeks, with medium change and fresh drug added twice a week. Cells were washed with PBS, trypsinized, and the relative percentage of RFP$^+$ cells was determined by flow cytometry analysis using a CytoFLEX S flow cytometer (Beckman Coulter). Data were visualized with FlowJo software (FlowJo).

## Flow cytometry to assess YAP reporter activity

HEK293A-GTIIC-GFP-Cas9 cells or corresponding gene edited cell derivatives were seeded in 6-well tissue culture plates at the same density and allowed to adhere overnight. The following day, cells were trypsinized, collected in cell culture medium and subjected to flow cytometry analysis using a CytoFLEX S flow cytometer (Beckman Coulter). Data were analyzed by FlowJo software (FlowJo).

## RHOA G-LISA activation assay

Cells were seeded in 6-well tissue culture plates at the same density and allowed to grow until about 50% of confluence. Cells were then lysed for RHOA G-LISA activation assay according to the manufacturer's instructions (Cytoskeleton Cat# BK124). Immunoblotting analysis was performed to assess the total RHOA protein level in the whole cell lysate.

## Protein extraction and immunoblotting

Cells were lysed in RIPA buffer (25 mM Tris-HCl pH 7.6, 150 mM NaCl, 1% NP-40, 1% sodium deoxycholate, 0.1% SDS) (Thermo Fisher Scientific #89901) supplemented with 100 x Protease Inhibitor Cocktail (Sigma #P8340), 100 x Phosphatase Inhibitor Cocktail (Thermo Fisher Scientific #78427) and 25 units/ml Benzonase Nuclease (Sigma #E8263). Lysate was sonicated using a Diagenode Bioruptor 300 (High setting, 30 s on, 30 s off, 10 cycles), followed by centrifugation at 13,000 rpm, 4°C, 10 min. Protein concentration was determined using the DC Protein Assay Kit (Bio-Rad #5000112) according to the manufacturer's instructions. Equal amount of proteins were resolved by SDS-PAGE and transferred to nitrocellulose membranes (Bio-Rad #1704159EDU) using a Trans-Blot Turbo Transfer System (Bio-Rad #1704150EDU) according to the manufacturer's instructions. Membranes were blocked for 1 hr at room temperature with 5% Blotting-Grade Blocker (Bio-Rad #1706404) in Tris Buffered Saline with Tween 20, pH 8.0 (TBST) (Sigma #T9039) and then incubated overnight at 4°C with primary antibodies diluted in 5% BSA (Akron Biotechnology #AK8917-0100). Membranes were washed with TBST, followed by incubation with horseradish peroxidase (HRP) conjugated secondary antibody diluted in 5% Blotting-Grade Blocker and visualization with Amersham ECL Western Blotting Detection Reagents (GE Healthcare #RPN2106) or SuperSignal West Pico PLUS Chemiluminescent Substrate (Thermo Fisher Scientific #34580) and Amersham Hyperfilm ECL (GE Healthcare #28906839).

## Antibodies

The following antibodies were used in this study (with dilution factor for immunoblotting): anti-PKN2 (#2612, 1:1,000), anti-YAP (#14074, 1:1,000), anti-YAP/TAZ (#8418, 1:1,000), anti-phospho-YAP (Ser127) (#4911, 1:1,000), anti-phospho-EGFR (Tyr1068) (#3777, 1:1,000), anti-EGFR (#4267, 1:20,000), anti-phospho-AKT (Ser473) (#4058, 1:1,000), anti-AKT (#9272, 1:5,000), anti-phospho-ERK1/2 (Thr202/Tyr204) (#9101, 1:1,000), anti-ERK1/2 (#9102, 1:5,000), anti-ARIH2/TRIAD1 (#13689, 1:1,000), anti-BTAF1 (#2637, 1:1,000), anti-GNAQ (#14373, 1:1,000), anti-HSP90 (#4877, 1:5,000), anti-ALDOA (#8060, 1:5,000), anti-METAP2 (#12547, 1:1,000), anti-GAPDH (#2118, 1:5,000), anti-RHOA (#2117, 1:1,000), anti-Cofilin (#5175, 1:1,000), anti-phospho-Cofilin (Ser3) (#3313, 1:1,000), from Cell Signaling Technology; anti-α-Tubulin (T6074, 1:20,000) and anti-β-Actin (A1978, 1:20,000) from Sigma; anti-GNB2 (ab81272, 1:1,000), anti-RIC8A (ab97808, 1:1,000), anti-USP22 (ab195289, 1:1,000), anti-CUL5 (ab184177, 1:1,000), anti-RNF7 (ab181986, 1:1,000), anti-PDCD10 (ab180706, 1:1,000), from Abcam; anti-KCTD5 (#15553–1-AP, 1:1,000), anti-PSAT1 (#10501–1-AP, 1:5,000), from

Proteintech; Goat Anti-Rabbit IgG Antibody, (H+L) HRP conjugate (#AP307P, 1:5,000), Goat Anti-Mouse IgG Antibody, (H+L) HRP conjugate (#AP308P, 1:5,000), from Millipore Sigma.

## Assessing nascent protein synthesis by AHA labeling

Isogenic cell lines used to assess nascent protein synthetic rate were grown in 60 mm dishes until ~70% confluent. Cells were treated with DMSO or 1 µM Erlotinib for 24 hr. Prior to labeling, cells were washed with PBS and then incubated at 37°C with methionine-free media (Life Technologies, A1451701) containing 2% FBS supplemented with DMSO or 1 µM Erlotinib for 1 hr. Medium was then replaced by methionine-free media containing 2% FBS supplemented with 50 µM Click-iT AHA (L-azidohomoalanine) (Life Technologies, C10102) and the labeling was performed by incubation at 37°C for 1 hr. Cells were washed three times with PBS and immediately lysed with 200 µl lysis buffer (50 mM Tris-HCl, pH 8.0, 1% SDS, supplemented with protease and phosphatase inhibitors and Benzonase at appropriate concentrations). Cells were scraped off the dish and transferred into 1.5 ml microcentrifuge tubes. Complete cell lysis was achieved by sonication (Diagenode Biorupter 300: high setting, 30 s on, 30 s off, 10 cycles), followed by centrifugation at 13,000 rpm, 4°C, 5 min. Protein concentration was determined using the DC Protein Assay Kit (Bio-Rad #5000112) according to the manufacturer's instructions. 80–100 µg of protein lysates were then subjected to Click-iT reaction for switching azido-modified nascent proteins to alkyne-biotin (Life Technologies, B10185) using the Click-iT Protein Reaction Buffer Kit (Life Technologies, C10276), followed by protein precipitation according to the manufacturer's protocol. Air dried protein samples were re-solubilized in 60 µl of 1% SDS in water with vortex followed by heating the samples at 99°C for 10 min. Solubilized protein samples were cleared by centrifugation at 13,000 rpm for 1 min to remove any insoluble material. Protein concentration was determined using the DC Protein Assay Kit, and 15 µg of proteins were resolved by Tris-Glycine SDS-PAGE gel. Biotinylated nascent proteins were subjected to immunoblotting using Streptavidin-HRP (Cell Signaling Technology #3999, 1:5,000).

## Assessing nascent protein synthesis of METAP2

To assess METAP2 protein synthesis, cells were treated as above with the exceptions that AHA labeling is for 3 hr and 150 µg of protein lysates were used for Click-iT reaction. After Click-iT reaction and protein precipitation, air dried protein samples were re-solubilized in 50 µl of 1% SDS in Triton-X lysis buffer (50 mM Tris-HCl, pH 7.5, 150 mM NaCl, 10% glycerol, 1% Triton X-100) with vortex followed by heating the samples at 99°C for 10 min. Solubilized protein samples were cleared by centrifugation at 13,000 rpm for 1 min to remove any insoluble material. Protein concentration was determined using the DC Protein Assay Kit. Samples were diluted with Triton-X lysis buffer to reduce the amount of SDS and same amount of proteins were then used for streptavidin magnetic beads (Thermo Fisher Scientific Cat# 88817) pulldown with incubation at 4°C for overnight. The following day, beads were collected with a magnetic stand and washed three times with Triton-X lysis buffer. After washing, beads were collected and re-suspended with 25 µl of SDS-PAGE reducing sample buffer followed by heating at 99°C for 10 min. The beads were then magnetically separated and the supernatant was saved for western blotting analysis.

## TUBE assay

In vivo poly-ubiquitination of METAP2 was evaluated by the TUBE assay. Briefly, cells were treated with 0.1 µM of bortezomib for overnight to enrich poly-ubiquitination of target protein. Cell lysates were prepared and equal amount of cell lysates were incubated with agarose-TUBE beads (LifeSensors Cat# UM402) per the manufacturer's instructions. Then the beads were washed three times with lysis buffer, and bound proteins were eluted in SDS-PAGE reducing sample buffer for immunoblotting analysis.

## Protein identification by mass spectrometry

Cell pellet was lysed with urea buffer: 8 M urea (Sigma, #U1250), 1% SDS (Promega, #V6551), 50 mM Tris (Sigma, #10708976001) pH 8.5, phosphatase inhibitor cocktail tablet PhosSTOP (Roche, #4906837001). Proteins were reduced with 5 mM dithiothreitol (DTT, Sigma, #D9779) 1 hr at room temperature and alkylated with 15 mM iodoacetamide (IAA, Sigma, #I6125) for 1 hr at room temperature in the dark. Proteins were then precipitated with chloroform/methanol to remove salt and

detergent. After dissolving the dry protein pellet with 8 M urea, 50 mM Tris pH 8.5; the proteins were digested overnight with trypsin (Promega, #V5072) after dilution to 2 M urea. The peptides were acidified to 1% TFA, desalted on SepPak C18 cartridges and eluted with 60% acetonitrile, 0.1% TFA. Dried peptides were resuspended in 0.1 M TEAB buffer, pH 8.5 and then labeled with TMT reagent (1:4; peptide:TMT label) (Thermo Fisher Scientific). The reaction was quenched with 0.5% TFA and the six samples were combined to a 1:1 ratio.

Mixed and labeled peptides were subjected to high-pH reversed-phase HPLC fractionation on an Agilent X-bridge C18 column (3.5 μm particles, 2.1 mm i.d., and 15 cm in length). Using an Agilent 1200 LC system, a 60 min linear gradient from 10% to 40% acetonitrile in 10 mM ammonium formate separated the peptide mixture into a total of 96 fractions, which were then consolidated into 24 fractions. The dried 24 fractions were reconstituted in 0.1% formic acid for LC-MS3 analysis.

Labeled peptides were loaded onto a 15 cm column packed in-house with ReproSil-Pur 120 C18-AQ 1.9 μM (75 μm inner diameter) in an EASY-nLC 1200 system. The peptides were separated using a 120 min gradient from 3% to 30% buffer B (80% acetonitrile in 0.1% formic acid) equilibrated with buffer A (0.1% formic acid) at a flow rate of 250 nl/min. Eluted TMT peptides were analyzed on an Orbitrap Fusion Lumos mass spectrometer (Thermo Fisher Scientific).

MS1 scans were acquired at resolution 120,000 with 350–1500 m/z scan range, AGC target $2 \times 10^5$, maximum injection time 50 ms. Then, MS2 precursors were isolated using the quadrupole (0.7 m/z window) with AGC $1 \times 10^4$ and maximum injection time 50 ms. Precursors were fragmented by CID at a normalized collision energy (NCE) of 35% and analyzed in the ion trap. Following MS2, synchronous precursor selection (SPS) MS3 scans were collected by using high energy collision-induced dissociation (HCD) and fragments were analyzed using the Orbitrap (NCE 65%, AGC target $1 \times 10^5$, maximum injection time 120 ms, resolution 60,000).

Protein identification and quantification were performed using Proteome Discoverer 2.1.0.81 with the SEQUEST algorithm and Uniprot human database (2014-01-31, 21568 protein sequences). Mass tolerance was set at 10 ppm for precursors and at 0.6 Da for fragment. Maximum of 3 missed cleavages were allowed. Methionine oxidation was set as dynamic modification; while TMT tags on peptide N termini/lysine residues and cysteine alkylation (+57.02146) were set as static modifications. The list of identified peptide spectrum matches (PSMs) was filtered to respect a 1% False Discovery Rate (FDR) after excluding PSMs with an average TMT reporter ion signal-to-noise value lower than 10 and a precursor interference level value higher 50%. The Student's test was applied to identify significantly changed protein abundances and adjusted p-values were calculated according to Benjamin and Hochberg. The final list of identified proteins was filtered to achieve a 5% protein FDR.

## RNA extraction, reverse transcription and quantitative RT-PCR

Total RNA was extracted using the RNeasy Plus Mini Kit (Qiagen #74134) and reverse transcribed with TaqMan Reverse Transcription Reagents (Applied Biosystems #N8080234) according to the manufacturer's instructions. The resulting cDNA products were diluted and subjected to quantitative real-time PCR (qPCR) reactions using TaqMan Gene Expression Assays (Applied Biosystems). Specifically, qPCR was performed in 10 μl reactions consisting of 0.5 μl TaqMan probe (Applied Biosystems), 5 μl TaqMan Fast Advanced Master Mix (Applied Biosystems #4444557) and 4.5 μl diluted cDNA template. Experiments were run on a ViiA 7 Real-Time PCR System (Applied Biosystems). The thermocycling conditions used were 20 s at 95°C, followed by 40 cycles of 1 s at 95°C and 20 s at 60°C. The threshold crossing value (Ct) was determined for each transcript and normalized to the housekeeping gene transcript (*GUSB*). The relative quantification of each mRNA species was assessed using the comparative ΔΔCt method.

TaqMan probes used in this study are as follows: *GUSB* (Hs00939627_m1), *ANKRD1* (Hs00173317_m1), *CTGF* (Hs00170014_m1), *CYR61* (Hs00155479_m1), *METAP2* (Hs00199152_m1), *ALDOA* (Hs00605108_g1), *PSAT1* (Hs00795278_mH).

## In vivo xenograft study

All animal work was performed in accordance with Novartis Animal Care and Use Committee (ACUC) regulations and guidelines. All animals were allowed to acclimate in the Novartis animal facility with access to food and water ad libitum for 3 days prior to manipulation. All cell lines were confirmed as mycoplasma- and rodent pathogens-negative (IMPACT VIII PCR Profile, IDEXX) before

implantation. Female athymic nude mice (nu/nu, Charles River Laboratories), 6–8 weeks old, were inoculated subcutaneously with 20 million cells suspended in 50% Hank's balanced salt solution + 50% phenol red-free Matrigel (BD Biosciences). Mice were enrolled in the study once tumors had reached approximately 200 mm$^3$ in size (day 13 post-implantation), and were randomly assigned to receive either vehicle or erlotinib (LC Laboratories; 10 mg/kg) (compound formulation 50% Dexolve-7 (Generic SBECD) and 50% of 0.1 M Tartaric Acid) once daily by oral gavage for the duration of the study. Animal body weights were recorded and tumors were measured twice weekly by calipering in two dimensions. Tumor volume was calculated using the following formula: tumor volume (mm$^3$) = $L$ x $W^2$/2, where $L$ is the longest length of the tumor and $W$ is the length of the tumor perpendicular to $L$.

## Statistical analysis

Unless otherwise noted, data are presented as the means ± SD. For all graphs, data are presented relative to their respective controls. Statistical analyses were performed using GraphPad Prism 8 (Graphpad software Inc). Significance was determined by a two-tailed Student's $t$ test or ordinary two-way ANOVA, denoted within each figure panel and respective figure legends. $p < 0.05$ was considered to be statistically significant. $*p < 0.05$, $**p < 0.01$, $***p < 0.001$, $****p < 0.0001$; ns, not significant.

## Data availability

The mass spectrometry proteomics data have been deposited to the ProteomeXchange Consortium via the PRIDE partner repository with the dataset identifier PXD014198. CRISPR-Cas9 screen data were summarized in *Supplementary file 1* and *Supplementary file 2*.

## Acknowledgements

We thank Lihe Zhang, Huili Zhai and Tomas Rejtar for technical assistance. We also thank members of the Discovery Biology II group in the Department of Chemical Biology and Therapeutics, NIBR, for helpful discussions.

## Additional information

### Competing interests

Hao Zeng, Johnny Castillo-Cabrera and Mika Manser, Bo Lu, Zinger Yang, Vaik Strande, Damien Begue, Raffaella Zamponi, Shumei Qiu, Frederic Sigoillot, Qiong Wang, Alicia Lindeman, John S Reece-Hoyes, Carsten Russ, Debora Bonenfant, Xiaomo Jiang, Youzhen Wang, Feng Cong: affiliated with Novartis Institutes for Biomedical Research. No other competing interests to declare.

### Funding

No external funding was received for this work.

### Author contributions

Hao Zeng, Conceptualization, Data curation, Formal analysis, Supervision, Validation, Investigation, Visualization, Methodology, Writing—original draft, Project administration, Writing—review and editing, Designed the study; Johnny Castillo-Cabrera, Mika Manser, Bo Lu, Validation, Investigation, Methodology; Zinger Yang, Data curation, Software, Formal analysis; Vaik Strande, Damien Begue, Shumei Qiu, Youzhen Wang, Investigation, Methodology; Raffaella Zamponi, Validation; Frederic Sigoillot, Data curation; Qiong Wang, Resources; Alicia Lindeman, Methodology; John S Reece-Hoyes, Resources, Methodology; Carsten Russ, Data curation, Methodology; Debora Bonenfant, Data curation, Visualization, Methodology; Xiaomo Jiang, Conceptualization; Feng Cong, Conceptualization, Resources, Supervision, Project administration, Writing—review and editing, Designed the study

## Author ORCIDs

Hao Zeng https://orcid.org/0000-0003-4967-9555
Zinger Yang https://orcid.org/0000-0001-8543-4841

## Ethics

Animal experimentation: All animal work was performed in accordance with Novartis Animal Care and Use Committee (ACUC) regulations and guidelines (reference number 120137). All animals were allowed to acclimate in the Novartis animal facility with access to food and water ad libitum for 3 days prior to manipulation. All cell lines were confirmed as mycoplasma- and rodent pathogens-negative (IMPACT VIII PCR Profile, IDEXX) before implantation.

## Decision letter and Author response

Decision letter https://doi.org/10.7554/eLife.50223.sa1
Author response https://doi.org/10.7554/eLife.50223.sa2

## Additional files

### Supplementary files

• Supplementary file 1. Selected hits from the erlotinib resistance CRISPR-Cas9 screen. A threshold of RSA $\leq -3$ and Q1 z-score $\leq -1$ generated a list of 122 genes whose loss sensitized HCC827 cells to erlotinib treatment. A threshold of RSA $\leq -3$ and Q3 z-score $\geq 1$ generated a list of 171 genes whose loss conferred resistance to erlotinib in HCC827 cells.

• Supplementary file 2. Individual sgRNAs and log2 fold change for selected hits. Individual sgRNA target sequences and their respective log2 fold change based on the comparison of sgRNA abundance in the erlotinib-treated versus DMSO-treated cell population were listed in this table.

• Supplementary file 3. Key resources table.

• Transparent reporting form

### Data availability

The mass spectrometry proteomics data have been deposited to the ProteomeXchange Consortium via the PRIDE partner repository with the dataset identifier PXD014198. CRISPR-Cas9 screen data were summarized in Supplementary file 1 and Supplementary file 2.

The following dataset was generated:

| Author(s) | Year | Dataset title | Dataset URL | Database and Identifier |
|---|---|---|---|---|
| Zeng H, Cabrera JC, Manser M, Lu B, Yang Z, Strande V, Begue D, Zamponi R, Qiu S, Sigoillot F, Wang Q, Lindeman A, Hoyes JR, Russ C, Bonenfant D, Jiang X, Wang Y, Cong F | 2019 | Genome-wide CRISPR screening reveals genetic modifiers of mutant EGFR dependence in NSCLC | http://proteomecentral.proteomexchange.org/cgi/GetDataset?ID=PXD014198 | Pride, PXD014198 |

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
