## [Decision Letter]

**Acceptance summary:**

In this manuscript, Cong and colleagues used genome-scale CRISPR-Cas9 in a pooled format to look for genes whose depletion synergized with EGFR inhibition. They found 35 genes, some of which have been reported in prior screens. They identified and characterized two novel candidates RIC8A and ARIH2. For RIC8A they concluded that this gene regulates YAP1 signaling promising survival while ARIH2 upregulated METAP2. Overall these studies were performed at a high level and provide potential useful information about EGFR signaling.

**Decision letter after peer review:**

Thank you for submitting your article "Genome-wide CRISPR screening reveals genetic modifiers of mutant EGFR dependence in NSCLC" for consideration by *eLife*. Your article has been reviewed by three peer reviewers, one of whom is a member of our Board of Reviewing Editors, and the evaluation has been overseen by Richard White as the Senior Editor. The reviewers have opted to remain anonymous.

The reviewers have discussed the reviews with one another and the Reviewing Editor has drafted this decision to help you prepare a revised submission.

Essential revisions:

The reviewers have made the following comments that need to be addressed in the revised manuscript.

1) Please provide more details about the screen. In particular in Figure 1 and screen design, it looks like there could have been significant bottlenecking with the high dose of erlotinib early on in the screen which could affect library diversity and subsequent results, but the authors do not report or show the extent of killing that occurred. What% of cells survived this initial pulse? it would be very helpful to have more information on what representation looked like after this step.

2) The reviewers would like to see rescue experiments for the RIC8 and ARIH2 and for the follow-up experiments to include a second sgRNA. In addition, the authors should state whether they determined the copy number of these genes since they have previously reported on artefacts from targeting genes in amplicons.

3) For the RIC8A experiments, further experiments are need to understand how the YAP1 pathway is regulated. Is this solely due to affecting phosphorylation and what kinases are involved. In general, the studies concerning mechanisms are less developed. For example, it would beneficial to show whether the RIC8A, ARIH2 and METAP2 are involved in regulating TKI sensitivity in EGFR wild type expressing NSCLC cell lines and normal cell lines should be examined.

4) For the experiments involving ARIH2, whether ARIH2 inhibits METAP2 translation or regulating METAP2 degradation should be further investigated. Also iIs METAP2 a downstream effector of EGFR? The relative expression level of ARIH2, RIC8A and MERAP2 in EGFR wild type and mutant expressing NSCLC cell lines or tumors should be compared.

5) Details and data supporting the formation of a drug-resistant state (Figure 1, Figure 3) are missing. Is the screen being performed in a cell state in which all of the cells would have become resistant? How long does resistance take to form, and what fraction of cells die before this state is reached? Drug resistance and re-sensitization are of strong interest, but these are important details and should be substantiated and shown clearly. Similarly for Figure 5D it would be nice to see a full curve for cell loss, etc during DTP formation.

---

## [Author Response]

Essential revisions:The reviewers have made the following comments that need to be addressed in the revised manuscript.1) Please provide more details about the screen. In particular in Figure 1 and screen design, it looks like there could have been significant bottlenecking with the high dose of erlotinib early on in the screen which could affect library diversity and subsequent results, but the authors do not report or show the extent of killing that occurred. What% of cells survived this initial pulse? it would be very helpful to have more information on what representation looked like after this step.

In Figure 1 and screen design (new Figure 1D), we added the concentration of erlotinib and days of treatment to provide more details about the screen. To make the rationale of our screen design more understandable to the general readers of the paper, we added some HCC827 cell characterization data before introducing our CRISPR screen in our revised manuscript (new Figure 1A-C). HCC827, a very commonly used EGFR-mutant NSCLC cell line, harbors ex19 del in EGFR and is highly sensitive to EGFR-TKI treatment. We first validated its high sensitivity to erlotinib by generating a dose-response curve upon 3 days’ drug exposure, with the IC50 25.6 ± 3.6 nM (new Figure 1A). However, with the clinically-relevant concentration (1 µM) or even higher concentrations, there are still about 30% of the cells that could survive this initial pulse (new Figure 1A). We further monitored the cell proliferation in the presence of DMSO control or 1 µM erlotinib in a 6-well tissue culture plate over a 30-day period using the IncuCyte instrument, and showed that the HCC827 cells were still able to proliferate at a low rate when exposed to high dose of erlotinib (new Figure 1B). Additionally, colony formation assay also confirmed the existence of a small fraction of viable cells after 9 days’ 1 µM erlotinib treatment, referred to as “drug-tolerant persister” (DTP) cells (new Figure 1C). Thereafter, the drug-tolerant cells commenced cell proliferation in the presence of drug, yielding colonies of cells referred to as “drug-tolerant expanded persister” (DTEP) cells or drug resistant cells (new Figure 1C). These data suggested that EGFR inhibition in the cultured cells mimics clinical observations of the incomplete response and/or innate resistance to EGFR-TKI treatment, allowing the assay window to screen for mediators of EGFR-TKI sensitivity that can either prevent or promote the development of drug resistance. Therefore we believe it is the beauty of the screen design with high dose of erlotinib, rather than being a bottleneck.

Regarding the library diversity and sgRNA representation, we have now performed additional data analysis to show more details of the screen results (new Figure 1—figure supplement 1A-C). Since we did not collect samples after a short erlotinib exposure (referred to as “initial pulse” by the reviewers), we analyzed data from the original library pool and the endpoint of the screen. As expected, deep sequencing 21 days after DMSO treatment revealed a significant reduction (left shift of the curve) in the diversity of sgRNAs (new Figure 1—figure supplement 1A), due to the depletion of sgRNAs targeting essential genes. Erlotinib treatment further resulted in the left shift of the curve (new Figure 1—figure supplement 1A), due to the depletion of sgRNAs targeting the synthetic lethal genes. After 21 days of erlotinib treatment, the sgRNA distribution was significantly different when compared with DMSO-treated cells (new Figure 1—figure supplement 1B-C), enabling the identification of sgRNAs targeting genes that regulate EGFR-TKI sensitivity, such as RIC8A and ARIH2 (new Figure 1—figure supplement 1C).

2) The reviewers would like to see rescue experiments for the RIC8 and ARIH2 and for the follow-up experiments to include a second sgRNA. In addition, the authors should state whether they determined the copy number of these genes since they have previously reported on artefacts from targeting genes in amplicons.

To perform the rescue experiments, we generated constructs expressing HA tagged Cas9/sgRNA-resistant RIC8A (HA-RIC8A-mut) or ARIH2 (HA-ARIH2-mut) and then performed colony formation rescue experiments. As shown in new Figure 3—figure supplement 1B, overexpression of HA-RIC8A-mut rescued the effect from RIC8A KO in colony formation assay. Similarly, overexpression of HA-ARIH2-mut rescued the effect from ARIH2 KO in colony formation assay (new Figure 5—figure supplement 1). For the follow-up experiments involving RIC8A and ARIH2 knockout, we included two independent sgRNAs for all the new experiments that we performed (see below in this rebuttal letter). We also determined the copy number of these genes by searching the database (https://cansar.icr.ac.uk/cansar/cell-lines/HCC-827/copy_number_variation/no%20signal/) and did not find gain of copy numbers for any of the genes studied in the current manuscript. In particular, the copy number of RIC8A or ARIH2 in HCC827 cell line is 2. We included this statement in the Discussion section of our revised manuscript.

3) For the RIC8A experiments, further experiments are need to understand how the YAP1 pathway is regulated. Is this solely due to affecting phosphorylation and what kinases are involved. In general, the studies concerning mechanisms are less developed. For example, it would beneficial to show whether the RIC8A, ARIH2 and METAP2 are involved in regulating TKI sensitivity in EGFR wild type expressing NSCLC cell lines and normal cell lines should be examined.

RIC8A functions as a biosynthetic chaperone and guanine nucleotide exchange factor (GEF) for a subset of Gα proteins, which is important for Gα-coupled GPCR signaling activation. It has been well established that GPCR signaling coupled with Gα proteins activates YAP signaling which involves Rho/Rac-mediated cytoskeleton remodeling and regulation of the activity of kinases upstream of YAP, such as MST and LATS, resulting in the de-phosphorylation of YAP (Yu et al., 2012; Ma et al., 2019). Our study, for the first time, demonstrated that RIC8A mediates EGFR-TKI sensitivity by regulating the YAP signaling. To dive deeper into the connection between RIC8A and YAP signaling, we speculated that RIC8A positively regulates YAP signaling via signaling through Gα-Rho/Rac axis. We first tested whether RHOA inhibition could confer synthetic lethality with EGFR inhibition. As shown in the new Figure 4—figure supplement 2A-C, knockout of RHOA decreased YAP-target gene expression and induced synthetic lethality with EGFR inhibition in HCC827 cells. Then we examined whether RHOA activity is reduced by RIC8A KO. Unfortunately, after many attempts, we observed little decrease in the active RHOA signal upon RIC8A KO using a RhoA G-LISA Activation Assay Kit (Cytoskeleton Cat.# BK124) (new Figure 4—figure supplement 2D-E; panel D shows the western blots using whole cell lysate; panel E shows the active RHOA signal determined by the kit). However, this could be due to the fast dynamics of RHOA activation-inactivation cycle, making it difficult to capture the real-time changes in RHOA activity upon RIC8A KO. In addition, *ARHGAP29*, encoding a Rho GTPase activating protein, was previously reported to be a YAP target gene (Qiao et al., 2017). We observed that RIC8A KO caused significant decrease in the expression of *ARHGAP29* (new Figure 4—figure supplement 2F), which could provide a negative-feedback mechanism to alleviate the decrease of RHOA activity resulting from RIC8A KO. Therefore, we believe our inability of detecting RHOA activity changes is most likely due to both technical reasons and the negative-feedback mechanism. Detection of active small GTPase is notoriously difficult, and this is well known in the field. Consistently, we observed a morphological alteration in HCC827 cells upon RIC8A KO (new Figure 4—figure supplement 2G) and the decrease of Cofilin phosphorylation that is downstream of the RHOA-ROCK signaling (new Figure 4—figure supplement 2H), indicating that RIC8A KO negatively impacted the output from RHOA activation. Moreover, treatment of Y-27632, the inhibitor of Rho-associated kinase ROCK, also induced synthetic lethality with EGFR inhibition in HCC827 cells (new Figure 4—figure supplement 2I-J). Taken together, these data suggested the RIC8A-Gα-RHOA-YAP signaling axis as a mediator of EGFR-TKI sensitivity in EGFR-mutant NSCLC cells. Admittedly, RIC8A might regulate YAP signaling through other effectors, and comprehensive understanding of the signaling between RIC8A and YAP warrants future characterizations, which is beyond the scope of the current study.

To assess whether RIC8A, ARIH2 and METAP2 are involved in regulating EGFR-TKI sensitivity in EGFR-WT NSCLC cell lines and normal cells, we selected three EGFR-WT NSCLC cell lines (A549, NCI-H1299 and NCI-H460) and one normal human bronchial epithelial cell line BEAS-2B. These cell lines were engineered to stably express Cas9 protein for the purpose of CRISPR-Cas9 mediated gene editing. We knocked out RIC8A using two independent sgRNAs in all four cell lines, and examined the sensitivity of control and RIC8A KO cells to EGFR-TKIs (erlotinib and gefitinib) by generating the dose-response curves as well as performing the colony formation assays. As shown in the new Figure 3—figure supplement 2A-P, loss of RIC8A had no effect on the EGFR-TKI sensitivity in EGFR-WT NSCLC cell lines or normal cells. Similar experiments were also done for ARIH2 with the observation that loss of ARIH2 had no effect on the EGFR-TKI sensitivity in EGFR-WT NSCLC cell lines or normal cells, although ARIH2 KO led to the increase of METAP2 protein abundance (new Figure 5—figure supplement 3A-P).

4) For the experiments involving ARIH2, whether ARIH2 inhibits METAP2 translation or regulating METAP2 degradation should be further investigated. Also iIs METAP2 a downstream effector of EGFR? The relative expression level of ARIH2, RIC8A and MERAP2 in EGFR wild type and mutant expressing NSCLC cell lines or tumors should be compared.

First, we tested whether ARIH2 regulates METAP2 degradation. We treated HCC827 cells with the proteasome inhibitor bortezomib and observed increase of METAP2 protein abundance (new Figure 6I), suggesting that METAP2 protein level is regulated by the proteasome-dependent degradation pathway. Moreover, bortezomib induced METAP2 protein level increase was only observed in control cells but not in ARIH2 KO cells (new Figure 6J), suggesting that proteasome-mediated METAP2 protein degradation is dependent on ARIH2. We further attempted to examine METAP2 ubiquitination in HCC827 cells. Unfortunately, we were not able to detect endogenous METAP2 protein ubiquitination even in the control cells (new Figure 6—figure supplement 1G), likely due to that the METAP2 ubiquitination level is too low to be detected or ARIH2 regulates METAP2 degradation indirectly.

Next, we tested whether ARIH2 regulates METAP2 protein translation. Specifically, we assessed de novo METAP2 protein synthesis in control or ARIH2 KO cells by L-azido-homoalanine (AHA) labeling followed by streptavidin pulldown. Consistent with our previous findings (Figure 6G-H), ARIH2 KO did not increase the global protein synthesis in the absence of EGFRi (new Figure 6K). However, we observed that knockout of ARIH2 increased METAP2 protein synthesis, suggesting that ARIH2 indeed regulates METAP2 protein translation. As a control, EGFR protein synthesis remained unchanged upon ARIH2 KO (new Figure 6K). Taken together, these new data demonstrated that ARIH2 is capable of regulating both protein translation and protein degradation of METAP2. Future studies are definitely required to gain a comprehensive picture of the regulation of METAP2 protein level by ARIH2.

To test whether METAP2 is a downstream effector of EGFR, we treated HCC827 cells with EGFRi erlotinib. As shown in new Figure 6—figure supplement 1E-F, erlotinib treatment did not alter the METAP2 protein level in both control and ARIH2 KO cells, indicating that the likelihood of METAP2 being a downstream effector of EGFR is low.

Lastly, as suggested by the reviewers, we examined the expression levels of RIC8A, ARIH2 and METAP2 in a panel of EGFR-mutant and EGFR-WT NSCLC cell lines (new Figure 6—figure supplement 1D). They are all ubiquitously expressed and their expression levels vary among different cell lines. We cannot draw any solid conclusion from this experiment though.

5) Details and data supporting the formation of a drug-resistant state (Figure 1, Figure 3) are missing. Is the screen being performed in a cell state in which all of the cells would have become resistant? How long does resistance take to form, and what fraction of cells die before this state is reached? Drug resistance and re-sensitization are of strong interest, but these are important details and should be substantiated and shown clearly. Similarly for Figure 5D it would be nice to see a full curve for cell loss, etc during DTP formation.

We have now added the details, particularly the days of DMSO or EGFR-TKI treatment, for all the colony formation assays presented in our manuscript. For questions regarding the screen and the formation of drug resistance, please refer to our response to point 1. To show the full growth curve during DTP formation, we monitored the proliferation of control or ARIH2 KO cells in the presence of DMSO control or 1 µM erlotinib over a 9-day period. As shown in the new Figure 5D, exposure of erlotinib resulted in growth arrest of both control and ARIH2 KO cells, whereas the ARIH2 KO cells exhibited higher proliferation rate when compared with control cells in the presence of erlotinib.